# Understanding and Improving Robustness of Vision Transformers through Patch-Based Negative Augmentation

## Abstract

We investigate the robustness of vision transformers (ViTs) through the lens of their special patch-based architectural structure, i.e., they process an image as a sequence of image patches. We find that ViTs are surprisingly insensitive to patch-based transformations, even when the transformation largely destroys the original semantics and makes the image unrecognizable by humans. This indicates that ViTs heavily use features that survived such transformations but are generally not indicative of the semantic class to humans. Further investigations show that these features are useful but *non-robust*, as ViTs trained on them can achieve high in-distribution accuracy, but break down under distribution shifts. From this understanding, we ask: can training the model to rely less on these features improve ViT robustness and out-of-distribution performance? We use the images transformed with our patch-based operations as negatively augmented views and offer losses to regularize the training away from using non-robust features. This is a *complementary* view to existing research that mostly focuses on augmenting inputs with semantic-preserving transformations to enforce models' invariance. We show that patch-based negative augmentation consistently improves robustness of ViTs across a wide set of ImageNet based robustness benchmarks. Furthermore, we find our patch-based negative augmentation are complementary to traditional (positive) data augmentation, and together boost the performance further. All the codes in this work will be open-sourced.

## 1 Introduction

Building vision models that are robust, i.e., that are highly accurate even on unexpected and out-of-distribution images, is increasingly a requirement to trusting vision models and a strong benchmark for progress in the field. Recently, Vision Transformers (ViTs, Dosovitskiy et al. (2021)) sparked great interest in the literature, as a radically new model architecture offering significant accuracy improvements and with hope of new robustness benefits. Over the past decade, there has been extensive work on understanding the robustness of convolution-based neural architectures, as the dominant design for visual tasks; researchers have explored adversarial robustness (Szegedy et al., 2013), domain generalization (Xiao et al., 2020; Khani & Liang, 2021), feature biases (Brendel & Bethge, 2019; Geirhos et al., 2018; Hermann et al., 2020). As a result, with the new promise of vision transformers, it is critical to understand *their* properties and in particular their robustness. Recent early studies (Naseer et al., 2021; Paul & Chen, 2021; Bhojanapalli et al., 2021) have found ViTs be more robust than ConvNets in some scenarios, with the hypothesis that the non-local attention based interactions enabled ViTs to capture more global and semantic features. In contrast, we add to this line of research showing a different side of the challenge: we find ViTs are still vulnerable to relying on non-robust features impeding out-of-distribution performance.

In this paper, we first demonstrate ViTs rely on specific non-robust features and then show how to reduce the reliance on these non-robust features, enabling improved out-of-distribution performance. To understand the robustness properties of ViTs, we start with the architectural traits of ViTs – ViTs operate on non-overlapping image patches and allow long range interaction between patches even in lower layers. It is hypothesized in recent studies (Naseer et al., 2021; Paul & Chen, 2021; Bhojanapalli et al., 2021) that the non-local attention based interactions contribute to better robustness

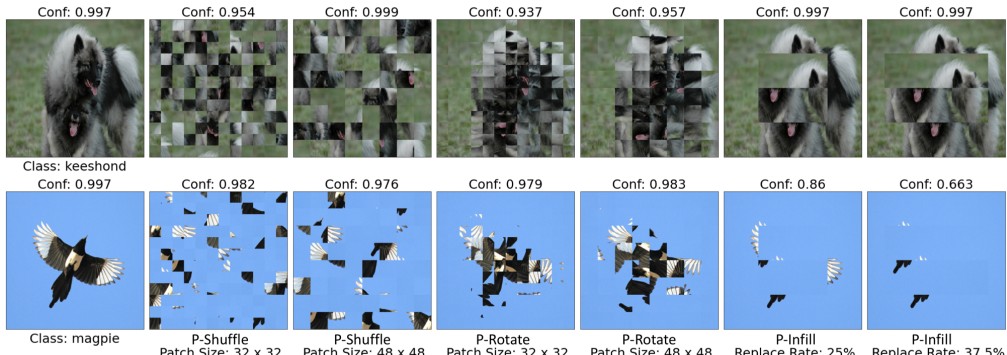

Figure 1: Patch-based transformations largely destroy images to be unrecognizable to humans whereas ViT recognizes them as the original class (e.g., keeshond or magpie) with high confidence. Visualization of patch-based transformations. On the top of each image, we display the predicted confidence score of ViT-B/16 pretrained on ImageNet-21k and finetuned on ImageNet-1k.

of ViTs than ConvNets. To study the ability of ViTs to integrate global semantics structures across patches, we design and apply patch-based image transformations, such as random patch rotation, shuffling, and background-infilling (Figure 1). Those transformations destroy the spatial relationship between patches and corrupted the global semantics, and the resultant images are often visually unrecognizable. However, we find that ViTs are surprisingly insensitive to these transformations and can make highly accurate predictions on these transformed images. This suggests that ViTs use features that survive such transformations but are generally not indicative of the semantic class to humans. Going one step further, we find that those features are useful but not robust, as ViTs trained on them achieved high in-distribution accuracy, but suffered significantly on robustness benchmarks.

With this understanding of ViTs' reliance on non-robust features captured by patch-based transformations, we still must answer: (a) how can we train ViTs to not rely on such features? and (b) will reducing reliance on such features meaningfully improve out-of-distribution performance and not sacrifice in-distribution accuracy? A majority of past robust training algorithms encourage the smoothness of model predictions on augmented images with semantic *preserving* transformations (Hendrycks et al., 2020b; Cubuk et al., 2019). However, the patch-based transformations deliberately destroy the semantic meaning and only leave non-robust features. Taking inspiration from recent research on generative modeling (Sinha et al., 2020), we propose a family of robust training algorithms based on *patch-based negative augmentations* that regularize the training from relying on non-robust features surviving patch-based transformations. Through extensive evaluation on a wide set of ImageNet-based benchmarks, we find that our methods consistently improve the robustness of the trained ViTs. Furthermore, our patch-based negative augmentation can be combined with the traditional (positive) data augmentation to boost the performance further. With this we get a more complete picture: training models both to be insensitive to spurious changes (as in positive augmentation) but also to not rely on non-robust features (as in negative augmentation) together can meaningfully improve robustness of ViTs.

Our key contributions are as follows:

- **Understanding Non-Robust Features in ViT:** We show that ViTs heavily rely on non-robust features surviving patch-based transformations but are not indicative of the semantic classes to humans.

- **Modeling:** We propose a set of patch-based operations as negatively augmented views, complementary to existing works that focus on semantic-preserving ("positive") augmentations, to regularize the training away from using these specific non-robust features;

- **Improved Robustness of ViT:** We show across a wide set of robustness benchmarks that our proposed negative augmented views can consistently improve ViT's robustness and complementary to "positive" augmentation.

## 2 PRELIMINARIES

**Vision Transformers** Vision transformers (Dosovitskiy et al., 2021) are a family of architectures adapted from Transformers in natural language processing (Vaswani et al., 2017), which di-

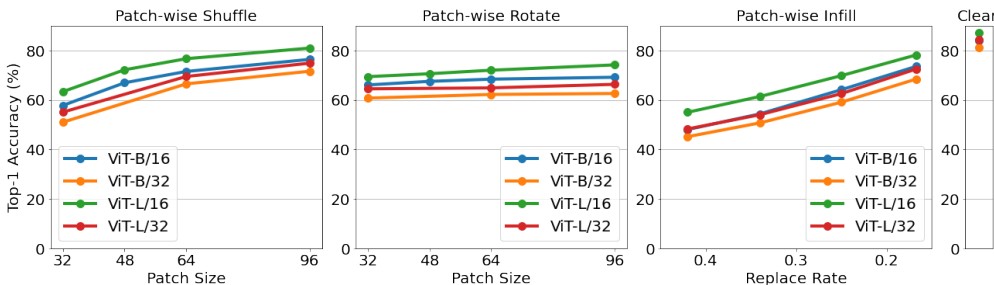

Figure 2: ViTs can rely on features surviving patch-based transformations to maintain a high accuracy, even after images have been heavily transformed to be largely unrecognizable. Top-1 accuracy of ViT models when tested on patch-based transformed images using the semantic class of the corresponding clean image as ground-truth. The test accuracy on ImageNet-1k validation set is shown on the right. All ViT models are pre-trained on ImageNet-21k and fine-tuned on ImageNet-1k.

rectly process visual tokens constructed from image patch embeddings. To construct visual tokens, ViT (Dosovitskiy et al., 2021) first splits an image into a grid of patches. Each patch is linearly projected into a hidden representation vector, and combined with a positional embedding. A learnable class token is also added. Transformers (Vaswani et al., 2017) are then directly applied on this set of visual and class token embeddings as if they are word embeddings in the original Transformer formulation. Finally, a linear projection of class token is used to calculate the class probability.

**Model Variants**   We consider ViT models pretrained on either ILSVRC-2012 ImageNet-1k, with ∼1.3 million images or ImageNet-21k, with ∼14 million images (Russakovsky et al., 2015). All models are fine-tuned on ImageNet-1k dataset. We adopt the notations used in (Dosovitskiy et al., 2021) to denote model size and input patch size. For example, ViT-B/16 denotes the "Base" model variant with input patch size $16 \times 16$.

**Robustness Benchmarks**   To evaluate models' robustness, we mainly focus on three ImageNet-based robustness benchmarks, ImageNet-A (Hendrycks et al., 2019), ImageNet-C (Hendrycks & Dietterich, 2019), ImageNet-R (Hendrycks et al., 2020a). Specifically, ImageNet-A contains challenging natural images from a distribution unlike ImageNet training distribution, which can easily fool models to make a misclassification. ImageNet-C consists of 19 types of corruptions that are frequently encountered in natural images and each corruption has 5 levels of severity. It is widely used to measure models' robustness under distributional shift. ImageNet-R is composed of images obtained by artistic rendition of ImageNet classes, e.g., cartoons, and is widely used to evaluate model's robustness on out-of-distribution data.

## 3   UNDERSTANDING ROBUSTNESS OF VISION TRANSFORMERS

Recent works (Naseer et al., 2021; Paul & Chen, 2021; Bhojanapalli et al., 2021) have shown that vision transformers achieve better robustness compared to standard convolutional networks. One explanation for vision transformers' stronger robustness is that the attention mechanism can capture better global structures. To investigate if ViT has successfully taken advantage of the long range interactions between patches, we design a series of patch-based transformations which significantly destroys the global structure of images. The patch-based transformations (see Fig. 1) are:

- **Patch-based Shuffle (P-Shuffle)**: we randomly shuffle the input image patches to change their positions[1].

- **Patch-based Rotate (P-Rotate)**: we randomly select a rotation degree from the set $\Omega = \{0°, 90°, 180°, 270°\}$ and rotate each image patch independently.

- **Patch-based Infill (P-Infill)**: we replace the image patches in the center region of an image with the patches on the image boundary[2].

---

[1] P-Shuffle is equivalent to shuffling the position embeddings.

[2] For example, given an image with size $384 \times 384$, input patch size is $16 \times 16$ and replace rate 0.25, we in total have 576 patches $\boldsymbol{x}_{i,j}$, where $i$ and $j$ denotes the row and column index and $1 \leq i, j \leq 24$. The patches in the center $\boldsymbol{x}_{m,n}, 7 \leq m, n \leq 18$ are replaced by the remaining patches.

Figure 3: Features preserved in patch-based transformations are useful but non-robust as training ViT on them impedes robustness. Top-1 Accuracy (%) on ImageNet-1k validation set and ImageNet robustness datasets: ImageNet-A, ImageNet-C, ImageNet-R. The baseline model is ViT-B/16 in (Dosovitskiy et al., 2021) trained on original images. Other models are trained on patch-based transformed images, e.g., "P-Shuffle" stands for a ViT-B/16 model trained on patch-based shuffled images. Numbers above the bars are either accuracy (e.g., ViT-B/16) or the *max* accuracy difference between each model family and the baseline ViT-B/16. The patch size in P-Shuffle and P-Rotate and replacement ratio in P-Infill is denoted by "ps" and "rr" respectively.

Each patch-based transformation is performed to a single image. We make sure the patch size of our patch-based transformation is a multiple of the input image patch of ViT so that the content within each patch is well-maintained. For P-infill, we use "replace rate" to denote the ratio of replaced patches in the center over the total number of patches in an image. Examples of transformed images are shown in Fig. 1 (see Appendix F for more examples). In most cases, it is challenging to recognize the semantic classes after those transformations.

***Do ViTs rely on features not indicative of the semantic classes to humans?*** To validate if ViTs behave similarly as humans on these patch-based transformed images, we *evaluate* ViT models (Dosovitskiy et al., 2021) on these patch-based transformed image. Specifically, we apply each patch-based transformation to ImageNet-1k validation set and report the test accuracy of each ViT on the transformed images. The test accuracy is computed by using the semantic class of the corresponding original image as the ground-truth. As shown in Figure 2, the accuracy achieved by ViTs are significantly higher than random guessing (0.1%). In addition, as shown in Figure 1, ViT gives these patch-based transformed images a very high-confident prediction even when the transformation largely destroys the semantics and make the image unrecognizable by humans [3]. This strongly indicates that ViT models heavily rely on the features that survive these transformations to make a prediction. However, these features are not indicative of the semantic class to humans.

***Do features preserved in patch-based transformations impede robustness?*** Taking one step further, we want to know if the features preserved by simple patch-based transformations, which are not indicative of the semantic class to humans, result in robustness issues. To this end, we train a vision transformer, e.g., ViT-B/16, on patch-based transformed images with original semantic class assigned as their ground-truth. Note that all the training images are patch-based transformed images. In this way, we force the model to fully exploit the features preserved in patch-based transformations. In addition, we reuse all the training details and hyperparameters in (Dosovitskiy et al., 2021) to make sure the "only" difference between our models and the baseline ViT-B/16 (Dosovitskiy et al., 2021) is the training images. Then, we test the model on ImageNet-1k validation set and three robustness benchmarks, ImageNet-A, ImageNet-C and ImageNet-R *without any transformation*.

First, we can observe that for the baseline ViT-B/16, compared to in-distribution accuracy, all the out-of-distribution accuracies have suffered from a significant drop (the 4 blue bars in Figure 3). This trend has been observed both for ViTs and convolution-based networks (Paul & Chen, 2021; Zhai et al., 2021). Second, if we compare the accuracy between the baseline model and models trained on patch-based transformations (i.e., the difference between the blue bar and one of the red/green/orange bars in Figure 3), we find that ViTs' in-distribution accuracy drops only slightly, but the robustness drop is significant when models are trained on these patch-based transformations. Take P-Shuffle as an example, the model trained on patch-based shuffled images can still achieve 79.1% accuracy on ImageNet-1k, only 5 percentage point (pp) drop in in-distribution accuracy. In contrast, the accuracy drop on robustness datasets is much more significant, e.g., 17pp on ImageNet-R. The deterioration rate in robustness is close to 50% of the baseline ViT-B/16. This strongly

---

[3]Similar patterns are observed when ViT models are pretrained on ImageNet-1k.

suggests that the features preserved in patch-based transformations are sufficient for high-accurate in-distribution prediction but are not robust under distributional shifts.

Taking the above results together, we conclude that even though ViTs are shown to be more robust than ConvNets in previous studies (Naseer et al., 2021; Paul & Chen, 2021), they still heavily rely on features that are not indicative of the semantic classes to humans. These features, captured by patch-based transformations, are ***useful*** but ***non-robust***, as ViTs trained on them achieve high in-distribution accuracy but suffer significantly on robustness benchmarks.

With this knowledge we now ask: *based on these understandings, can we train ViTs to not rely on such non-robust features? And if we do, will it improve their robustness?*

## 4 IMPROVING THE ROBUSTNESS OF VISION TRANSFORMERS

Based on the key observations that the patch-based transformations encode features that contribute to the non-robustness of ViTs, we propose a *negative augmentation* procedure to regularize ViTs from relying on such features. To this end, we use the images transformed with our patch-based operations as negatively augmented views. Then we design negative loss regularizations to prevent the model from using those non-robust features preserved in patch-based transformations.

Specifically, given a clean image $x$, we generate its negative view, denoted as $\tilde{x}$, by applying a patch-based transformation to $x$. We call it *negative* augmentation, in contrast with the standard (positive) augmentation that are semantic preserving. Let $\mathcal{L}_{ce}(\mathbb{B}; \boldsymbol{\theta}) = -\frac{1}{|\mathbb{B}|} \sum_{(x,y) \in \mathbb{B}} y \log \mathrm{softmax}(f(x; \boldsymbol{\theta}))$ represent the cross-entropy loss function used to train a vision transformer with parameters $\boldsymbol{\theta}$, where $\mathbb{B}$ is a minibatch of clean examples, and $y$ denotes the ground-truth label. The loss on negative views $\mathcal{L}_{neg}(\mathbb{B}, \tilde{\mathbb{B}}; \boldsymbol{\theta})$ can be easily added to the cross-entropy loss $\mathcal{L}_{ce}(\mathbb{B}; \boldsymbol{\theta})$ via

$$\mathcal{L}_{ce}(\mathbb{B}; \boldsymbol{\theta}) + \lambda \cdot \mathcal{L}_{neg}(\mathbb{B}, \tilde{\mathbb{B}}; \boldsymbol{\theta}), \tag{1}$$

where $\lambda$ is a coefficient balancing the importance between clean training data as well as patch-based negative augmentation. Below, we introduce three different losses on negative views to leverage patch-based negative augmentation through label, logit and representation space respectively.

**Label space: uniform loss** Many existing data augmentation techniques (Cubuk et al., 2019; 2020; Hendrycks et al., 2020b) use one-hot labels for *semantic-preserving* augmented data to enforce the invariance of the model prediction. In contrast, the semantic classes of our generated patch-based negative augmented data are visually unrecognizable, as shown in Figure 1. Therefore, we propose to use uniform labels instead for those negative augmentations. Specifically, the loss function on negative views that we optimize at each training step can be formulated as:

$$\mathcal{L}_{neg}(\mathbb{B}, \tilde{\mathbb{B}}; \boldsymbol{\theta}) = -\frac{1}{|\tilde{\mathbb{B}}|} \sum_{(\tilde{x}, \tilde{y}) \in \tilde{\mathbb{B}}} \tilde{y} \log \mathrm{softmax}(f(\tilde{x}; \boldsymbol{\theta})), \tag{2}$$

where $\tilde{y}$ denotes the uniform distribution: $\tilde{y}_k = \frac{1}{K}$ where $K$ is the total number of classes. $f(x; \boldsymbol{\theta})$ denotes the function mapping the input image into the logit space.

**Logit space: $\ell_2$ Loss** An alternative to pre-assuming labels for negative augmentation is to add the constraints on the logit space (or the space of predicted probability). Inspired by existing work (Kannan et al., 2018; Zhang et al., 2019; Hendrycks et al., 2020b) which provides an extra regularization term encouraging similar logits between clean and "positive" augmented counterparts, we instead encourage the logits of clean examples and their corresponding negative augmentations to be far away. In this way, we prevent the model from relying on the non-robust features preserved in negative views. Specifically, we maximize the $\ell_2$ distance between the predicted probability of clean examples and their corresponding negative views. The loss on negative views, therefore, can be formulated as:

$$\mathcal{L}_{neg}(\mathbb{B}, \tilde{\mathbb{B}}; \boldsymbol{\theta}) = -\frac{1}{|\tilde{\mathbb{B}}|} \sum_{x \in \mathbb{B}, \tilde{x} \in \tilde{\mathbb{B}}} \|\mathrm{softmax}(f(x; \boldsymbol{\theta})) - \mathrm{softmax}(f(\tilde{x}; \boldsymbol{\theta}))\|_2. \tag{3}$$

Here the $\ell_2$ distance is computed over the predicted probability rather than the logits $f(x; \boldsymbol{\theta})$ because empirically we observe that maximizing the difference of logits can cause numerical instability.

**Representation space: contrastive loss** Lastly, we propose to use a contrastive loss (Oord et al., 2018; Chen et al., 2020a; Khosla et al., 2020) to regularize the training away from using non-robust features. For an example $x_i \in \mathbb{B}$, we create a positive set $\mathbb{P}_i \equiv \{x_j \in \mathbb{B}\backslash\{x_i\}|y_j = y_i\}$ with all the examples in the minibatch $\mathbb{B}$ sharing the same class as $x_i$. The anchor $x_i$ is excluded from its

positive set $\mathbb{P}_i$. Next, we can generate the negative set composed of two types of negative examples: 1) all the examples in the minibatch $\mathbb{B}$ with a different class as $\boldsymbol{x}_i$, 2) the patch-based negatively transformed images $\tilde{\boldsymbol{x}} \in \tilde{\mathbb{B}}$. For each anchor $\boldsymbol{x}_i$, we can in total have $2|\mathbb{B}| - |\mathbb{P}_i| - 1$ negative pairs, where $|\mathbb{B}|$ is the batch size and $|\mathbb{P}_i|$ is the cardinality of the positive set $\mathbb{P}_i$. Let the candidate set $\mathbb{Q}_i \equiv \tilde{\mathbb{B}} \cup \mathbb{B} \backslash \{\boldsymbol{x}_i\}$, the loss function can be expressed as:

$$\mathcal{L}_{neg}(\mathbb{B}, \tilde{\mathbb{B}}; \boldsymbol{\theta}) = -\frac{1}{|\mathbb{B}|} \sum_{\boldsymbol{x}_i \in \mathbb{B}} \frac{1}{|\mathbb{P}_i|} \sum_{\boldsymbol{x}_j \in \mathbb{P}_i} \log \frac{\exp(\mathrm{sim}(\boldsymbol{x}_i, \boldsymbol{x}_j)/\tau)}{\sum_{\boldsymbol{x}_k \in \mathbb{Q}_i} \exp(\mathrm{sim}(\boldsymbol{x}_i, \boldsymbol{x}_k)/\tau)}, \tag{4}$$

where $\tau$ is the temperature and $\mathrm{sim}(\boldsymbol{x}_i, \boldsymbol{x}_j) = \frac{g(\boldsymbol{x}_i;\boldsymbol{\theta})^\mathsf{T} \cdot g(\boldsymbol{x}_j;\boldsymbol{\theta})}{\|g(\boldsymbol{x}_i;\boldsymbol{\theta})\|\|g(\boldsymbol{x}_j;\boldsymbol{\theta})\|}$ computes the cosine similarity between $g(\boldsymbol{x}_i;\boldsymbol{\theta})$ and $g(\boldsymbol{x}_j;\boldsymbol{\theta})$, and $g(\boldsymbol{x};\boldsymbol{\theta})$ denotes the representation learned by the penultimate layer of the classifier. We do not use a learnable projection head[4] as in contrastive representation learning (Chen et al., 2020a;b). Therefore, no extra network parameters are used for our proposed method and the improvement of robustness can be mainly attributed to patch-based negative augmentations.

When the batch size is larger than the number of classes (which is the case for our ImageNet-1K experiments), it is easy to find positive examples from the same class in a mini-batch, so we only extend candidate set $\mathbb{Q}_i$ with our proposed patch-based negative data augmentations. When the batch size is far smaller than the number of classes (which is the case for our ImageNet-21K experiments), it can be difficult to find two examples from the same classes. Similar to (Chen et al., 2020a; Khosla et al., 2020), we generate another "positive" view for each image using common data augmentation (e.g., random cropping) so that we can make sure there is at least one positive pair from the same class. Denoting the set of positively augmented data as $\mathbb{B}^+$, the modified positive set is $\mathbb{P}_i \equiv \{\boldsymbol{x}_j \in \mathbb{B}^+ | \boldsymbol{y}_j = \boldsymbol{y}_i\}$, the candidate set $\mathbb{Q}_i$ is now $\tilde{\mathbb{B}} \cup \mathbb{B}^+$ instead of $\tilde{\mathbb{B}} \cup \mathbb{B} \backslash \{\boldsymbol{x}_i\}$.

## 5 EXPERIMENTS

**Experimental setup** We follow Dosovitskiy et al. (2021) to first pre-train all the models with image size $224 \times 224$ and then fine-tune the models with a higher resolution $384 \times 384$. We reuse all their training hyper-parameters, including batch size, weight decay, and training epochs (see Appendix A for details). For the two extra hyperparameters in our algorithms, the loss coefficient $\lambda$ in Eqn. 1 and the temperature $\tau$ in Eqn. 4 in contrastive loss, we sweep them from the set $\{0.5, 1, 1.5\}$ and $\{0.1, 0.5\}$ respectively and choose the model with the best hold-out validation performance. Please refer to Appendix B for the chosen hyperparameters for each model. We make sure our proposed models and our implemented baselines are trained with exactly the same settings for fair comparison. The top-1 accuracy of the fine-tuned models are reported on ImageNet-1k validation set as well as three robustness benchmarks, ImageNet-A (Hendrycks et al., 2019), ImageNet-C (Hendrycks & Dietterich, 2019) and ImageNet-R (Hendrycks et al., 2020a). For ImageNet-C, the reported accuracy is averaged over 19 corruptions types and 5 different corruption severities.

### 5.1 EFFECTIVE AND COMPLEMENTARY TO "POSITIVE" DATA AUGMENTATION

**Effective in improving robustness** First, we apply our proposed patch-based transformations to a ViT-B/16 model pre-trained and fine-tuned on ImageNet-1k. The extra loss regularization on negative views is used in both pre-training and fine-tuning stages to prevent the model from learning non-robust features preserved in patch-based transformations. We use "Transformation / Regularization" to denote a pair of patch-based negative augmentation and loss regularization. For examples, "P-Rotate / Uniform" means that we use P-Rotate to generate the negative views and use uniform loss to regularize the training. We display the results in Table 1, where we can clearly see that our proposed patch-based negative augmentation effectively improves the in-distribution test accuracy *and* the out-of-distribution robustness across all ImageNet-based benchmarks. We observe that all three loss regularizations effectively leverage the negative views to regularize the training away from using non-robust features, while the contrastive loss works the best.

**Complementary to traditional ("positive") data augmentation** To investigate if our proposed patch-based negative augmentation is complementary to traditional ("positive") data augmentation, we apply our patch-based negative transformation on top of traditional data augmentation: Rand-Augment (Cubuk et al., 2020), which is widely used in vision transformers (Touvron et al., 2021;

---

[4]We did not experiment if an extra projection head can push the result further as it is not our main focus but we encourage interested readers to validate if it is true or not.

Table 1: Top-1 accuracies for ViT-B/16 pre-trained and fine-tuned on ImageNet-1k with or without the proposed negative augmentation.

| Model | ImageNet-1k | ImageNet-A | ImageNet-C | ImageNet-R |
|---|---|---|---|---|
| ViT-B/16 (Dosovitskiy et al., 2021) | 77.6 | 6.7 | 50.8 | 20.3 |
| + P-Rotate / Uniform | 78.2 (+0.6) | 7.0 (+0.3) | 52.4 (+1.6) | 21.4 (+1.1) |
| + P-Rotate / L2 | 77.8 (+0.2) | 6.7 (+0.0) | 51.6 (+0.8) | 21.0 (+0.7) |
| + P-Rotate / Contrastive | 78.9 (+1.3) | 8.6 (+1.9) | 54.1 (+3.3) | 23.6 (+3.3) |

Table 2: Top-1 accuracies for ViT-B/16 pre-trained and fine-tuned on ImageNet-1k using Rand-Augment (Cubuk et al., 2020) or AugMix (Hendrycks et al., 2020b). The proposed negative augmentation is added on top of either positive augmentation. See Table 8 in Appendix for a **full** table with three losses for each patch-based transformation. Patch-based negative augmentation is complementary to "positive" data augmentation.

| Model | ImageNet-1k | ImageNet-A | ImageNet-C | ImageNet-R |
|---|---|---|---|---|
| Rand-Augment (Cubuk et al., 2020) | 79.1 | 7.2 | 55.2 | 23.0 |
| + P-Rotate / L2 | 79.1 (+0.0) | 7.9 (+0.7) | 56.7 (+1.5) | 23.8 (+0.8) |
| + P-Infill / Uniform | 79.2 (+0.1) | 7.8 (+0.6) | 56.4 (+1.2) | 24.0 (+1.0) |
| + P-Rotate / Contrastive | 79.9 (+0.8) | 9.4 (+2.2) | 58.4 (+3.2) | 25.4 (+2.4) |
| + P-Infill / Contrastive | 79.9 (+0.8) | 9.3 (+2.1) | 57.9 (+2.7) | 25.0 (+2.0) |
| AugMix (Hendrycks et al., 2020b) | 78.8 | 7.7 | 57.8 | 24.9 |
| + P-Rotate / L2 | 79.0 (+0.2) | 8.3 (+0.6) | 58.8 (+1.0) | 26.0 (+1.1) |
| + P-Infill / Uniform | 79.3 (+0.5) | 8.3 (+0.6) | 58.4 (+0.6) | 25.7 (+0.8) |
| + P-Rotate / Contrastive | 79.6 (+0.8) | 9.8 (+2.1) | 60.0 (+2.2) | 27.5 (+2.6) |
| + P-Infill / Contrastive | 79.6 (+0.8) | 9.9 (+2.2) | 60.3 (+2.5) | 27.3 (+2.4) |

Mao et al., 2021), and AugMix (Hendrycks et al., 2020b), which is specifically proposed to improve models' robustness under distributional shift. Crucially, we follow (Hendrycks et al., 2020b) to exclude transformations used in "positive" data augmentation which overlap with corruption types in ImageNet-C (Hendrycks & Dietterich, 2019). Therefore, the set of transformations used in Rand-Augment and AugMix is disjoint with the corruptions in ImageNet-C.

When we combine "negative" and "positive" augmentation, the cross-entropy loss $\mathcal{L}_{ce}(\mathbb{B}^+; \boldsymbol{\theta})$ in Eqn. 1 is computed over "positive" examples $\mathbb{B}^+ = \{\boldsymbol{x}_1^+, \cdots, \boldsymbol{x}_N^+\}$ using either Rand-Augment or AugMix. Meanwhile, the loss regularization on negative views in Eqn. 1 is computed over negatively transformed version of $\boldsymbol{x}^+$. That is: for $\forall \boldsymbol{x}^+ \in \mathbb{B}^+$, we apply our patch-based negative transformation to obtain its negative version and then use the negative example to compute the loss regularization $\mathcal{L}_{neg}$. The positive data augmentation is only used in pre-training stage as we observe it is slightly better than using them for both stages (Please see more detailed discussion in Appendix D) and Steiner et al. (2021) have the similar observation. Instead, we apply our negative augmentation in both stages, as it is the best design choice as discussed in Section 5.3.

As shown in Table 2, we see that when our patch-based negative augmentations are applied to either Rand-Augment or AugMix, we can consistently improve the robustness of vision transformers across all three robustness benchmarks (please refer to Table 8 in Appendix for a full table with three losses for each patch-based transformation). This is particularly noteworthy as both Rand-Agument and AugMix are already designed to significantly improve the robustness of vision models. Yet, we see that patch-based negative augmentation provides *further* robustness benefits. This suggests that robustness of vision models was not adequately addressed by "positive" data augmentation and that patch-based negative augmentation is complementary to these traditional approaches.

## 5.2 ROBUSTNESS IMPROVEMENTS EVEN UNDER LARGER PRE-TRAINING DATASETS

Considering that larger training data can significantly improve models' robustness and achieve state-of-the-art performance, we further investigate if our proposed method can scale up to larger datasets and continues to be necessary and valuable. To this end, we test if our proposed patch-based negative augmentation still helps robustness even when models are pre-trained on ImageNet-21k (10x larger than ImageNet-1k). Since we follow (Dosovitskiy et al., 2021) and use a batch size of 4096 in the pretraining stage, which is much less than the 21K classes in ImageNet-21k, it is unlikely to

Table 3: Top-1 accuracies of ViT-B/16 pretrained on ImageNet-21k and finetuned on ImageNet-1k. Patch-based negative augmentation is helpful even with large-scale pretraining.

| Model | ImageNet-1k | ImageNet-A | ImageNet-C | ImageNet-R |
|---|---|---|---|---|
| ViT-B/16 (Dosovitskiy et al., 2021) | 84.1 | 26.7 | 65.2 | 37.9 |
| Rand-Augment (Cubuk et al., 2020) | 84.4 | 28.7 | 67.2 | 38.7 |
| + P-Shuffle / Uniform | **84.5** (+0.1) | 29.9 (+1.2) | 67.7 (+0.5) | 38.9 (+0.2) |
| + P-Shuffle / L2 | **84.5** (+0.1) | 29.7 (+1.0) | 68.0 (+0.8) | **39.6** (+0.9) |
| + P-Shuffle / Contrastive | 84.3 (-0.1) | **30.8** (+2.1) | **68.1** (+0.9) | 38.6 (-0.1) |

Table 4: Effect of patch-based negative augmentation in pre-training and fine-tuning stages. Top-1 accuracies of ViT-B/16 pretrained and fine-tuned on ImageNet-1k. Under 'Stage' we denote which training stage patch-based negative augmentation is used.

| Model | Stage | ImageNet-1k | ImageNet-A | ImageNet-C | ImageNet-R |
|---|---|---|---|---|---|
| AugMix (Hendrycks et al., 2020b) | - | 78.8 | 7.7 | 57.8 | 24.9 |
| + P-Shuffle / Contrastive | Fine-tune | 79.2 (+0.4) | 8.3 (+0.6) | 58.4 (+0.6) | 25.9 (+1.0) |
| + P-Shuffle / Contrastive | Pre-train | 79.4 (+0.6) | 8.7 (+1.0) | 59.5 (+1.7) | 26.7 (+1.8) |
| + P-Shuffle / Contrastive | Both | **79.6** (+0.8) | **9.0** (+1.3) | **60.1** (+2.3) | **27.3** (+2.4) |

have multiple images of the same class in a mini-batch during pretraining. As mentioned above, we address this issue by augmenting each image one more time, generating another positive view of the same image, to make sure there is at least two examples from the same class in the mini-batch.

We use P-Shuffle as an example to generate negative views and display the results in Table 3 with negative augmentation in both pre-training and fine-tuning stages. We can clearly see that even when we greatly increase the size of pre-training dataset (i.e., ImageNet-21k is 10x larger than ImageNet-1k), our proposed patch-based negative augmentation can still further improve the robustness of ViT. This demonstrates that our approach is valuable at scale and improves models' robustness from an angle orthogonal to larger training data.

## 5.3 PRE-TRAINING VS. FINE-TUNING

We further disentangle the effect of patch-based negative data augmentation in pre-training and fine-tuning. Take P-Shuffle as an example, we design experiments to apply negative augmentation 1) only at the fine-tuning stage, 2) only at the pre-training stage, and 3) at both stages. As shown in Table 4, compared to the baselines, patch-based negative augmentation can effectively help improve robustness in both stages, and its effect in pre-training is slightly larger than in fine-tuning. Finally, we found using negative augmentation in both stages during training yields the largest gain. Please refer to Table 9 in Appendix for more settings, where the same conclusion holds.

## 5.4 UNDERSTANDING THE EFFECTS OF PATCH-BASED NEGATIVE AUGMENTATION

**Does ViT become more robust w.r.t. transformed images?** We further evaluate ViTs trained with our robust training algorithms on the patch-based transformed images. We found all three losses on negative views can successfully reduce the prediction accuracy of ViTs to be close to random guess (0.1%) with the original semantic classes as the ground-truth. In other words, our robust training algorithms make ViTs behave similarly as humans on those patch-based transformed images.

**Are texture biases contributing to non-robust features?** Geirhos et al. (2018) observed that unlike humans, CNNs rely on more local information (e.g., texture) rather than more global information (e.g., shape) to make a classification. Since our patch-based transformations largely destroy the global structure (e.g., shape), we want to investigate if the non-robust features surviving patch-based transformation overlap with local texture biases. To this end, we evaluate ViT-B/16 trained on patch-based transformations on Conflict Stimuli benchmark (Geirhos et al., 2018), and we see that ViTs trained *only* on patch-based transformation have a 4.9pp to 31.1pp increase on texture bias (Figure 4 in Appendix). This suggests that the useful but non-robust features preserved in patch-based transformation are indeed overlapped with the local texture bias. In addition, using our patch-negative augmentation can also to some extent reduce models' reliance on local texture bias, e.g., we decrease the texture accuracy from 71.7% to 62.2% for ViT-B/16 (Table 13 in Appendix).

### 5.5 ABLATION STUDY

**Sensitivity analysis** We test the sensitivity of our patch-based negative augmentation to various patch sizes in P-Shuffle and P-Rotate, and different replace rates in P-Infill. We find that P-Shuffle and P-Rotate are insensitive to patch sizes from $\{16, 32, 48, 64, 96\}$ for ViT-B/16, and P-Infill is robust to replace rates ranging from 1/3 to 1/2. The accuracy difference is smaller than 0.5% on ImageNet-1k as well as ImageNet-A and ImageNet-R. Therefore, we use the same parameter for all the settings investigated in this work (see Table 7 and Appendix B for details).

**Double batch-size of baselines** As we use the negative augmented view per example, the effective batch size is doubled compared to the vanilla ViT-B/16 trained with only cross-entropy loss. Therefore, we further investigate if the robustness improvement is a result from a larger batch size. When we increase the batch size from 4096 to 8192 in pre-training while keeping the same 300 training epochs, it decreases the in-distribution accuracy to 76.0% on ImageNet-1k as well as the accuracy on robustness benchmarks, e.g., ImageNet-R from 20.3% to 19.3%. Hence we conclude the robustness improvement is from the negative data augmentation we applied.

## 6 RELATED WORK

Vision transformers (Dosovitskiy et al., 2021; Touvron et al., 2021) are a family of Transformer models (Vaswani et al., 2017) that directly process visual tokens constructed from image patch embedding. Unlike convolutional neural networks (LeCun et al., 1989; Krizhevsky et al., 2012; He et al., 2016) that assume locality and translation invariance in their architectures, vision transformers have no such assumptions and are able to exchange information globally, thus having less inductive bias about the input image data. The significant difference in architectures raises questions about their robustness properties. A few recent studies find pretrained vision transformers are at least as robust as the ResNet counterparts (Bhojanapalli et al., 2021), and possibly more robust (Naseer et al., 2021; Paul & Chen, 2021). Our work studies a specific aspect of robustness pertaining patch-based visual tokens in ViT, and show it may lead to a generalization gap. Different from (Naseer et al., 2021) which also shows ViTs are insensitive to patch operations such as shuffle and occlusion, we further propose a mitigation strategy to increase robustness of patch-based architectures.

Data augmentation is widely used in computer vision models to improve model performance (Howard, 2013; Szegedy et al., 2015; Cubuk et al., 2020; 2019; Hendrycks et al., 2020b). It has been shown that data augmentation benefits vision transformers more than convolutional networks for relatively small scaled datasets (Touvron et al., 2021). However, most of the existing data augmentations are "positive" in the sense they assume the class semantic being preserved after the transformation. In this work, we explore "negative" data augmentation operations based on patches, where we encourage the representations of transformed example to be *different* from the original ones. Most related to our work in this direction is the work of Sinha et al. (2020). Although the concept of negative augmentation was proposed in their work, they only apply it for generative and unsupervised modeling. In contrast, our work focuses on discriminative and supervised modeling, and demonstrate how such negative examples can reveal specific robustness issues and such augmentation approaches can directly mitigate them, offering robustness improvements under large-scale pretraining settings.

## 7 CONCLUSION

Through this research we have found concrete examples of ViTs relying on non-robust features for predictions and shown that this reliance is limiting robustness and out-of-distribution performance. We believe this opens multiple exciting new lines of research. First, we believe that the methodological approach developed here is a valuable recipe for further progress. Through finding patch-wise, semantic-destroying transformations that ViTs are insensitive to we can identify when models rely on non-robust features, and through incorporating them as negative augmentations during training we can meaningfully reduce reliance on such features. Second, we believe this shows the potential for further improving the robustness of ViTs. Through training the model to use such non-robust features less, we have seen we can significantly improve the out-of-distribution performance of ViTs, without harming in-distribution accuracy! While we have identified multiple such non-robust features in ViTs, we believe that discovering and addressing more provides an avenue for valuable, on-going improvement of out-of-distribution performance. Taken together, we believe this is a promising direction for continued progress toward robust vision transformers and vision models in general.

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

## A  Training Details

We follow (Dosovitskiy et al., 2021) to train each model using Adam (Kingma & Ba, 2015) optimizer with $\beta_1 = 0.9$, $\beta_2 = 0.999$ for pre-training and SGD with momentum for fine-tuning. The batch size is set to be 4096 for pre-training and 512 for fine-tuning. All models are trained with 300 epochs on ImageNet-1k and 90 epochs on ImageNet-21k in the pre-training stage. In the fine-tuning stage, all models are trained with 20k steps except the models pretrained from ImageNet-1k without Rand-Augment (Cubuk et al., 2020) or Augmix (Hendrycks et al., 2020b), which we train them with 8k steps. The learning rate warm-up is set to be 10k steps. Dropout is used for both pre-training and fine-tuning with dropout rate 0.1. If the training dataset is ImageNet-1k, we additionally apply gradient clipping at global norm 1.

Table 5: Training details following (Dosovitskiy et al., 2021).

| Pre-train Dataset | Stage | Base LR | LR Decay | Weight Decay | Label Smoothing |
|---|---|---|---|---|---|
| ImageNet-1K | Pre-train | $3 \cdot 10^{-3}$ | 'cosine' | None | $10^{-4}$ |
| ImageNet-21k | Pre-train | $10^{-3}$ | 'linear' | 0.03 | $10^{-4}$ |
| ImageNet-1K | Fine-tune | 0.01 | 'cosine' | None | None |
| ImageNet-21K | Fine-tune | 0.03 | 'cosine' | None | None |

Table 6: Models using a different hyperparameter $\lambda$ than the default value (1.5).

| Model | Pre-train Dataset | Training stage | Hyperparameter $\lambda$ |
|---|---|---|---|
| Rand-Augment + P-Shuffle / Uniform | ImageNet-1k | Pre-train | 1.0 |
| Rand-Augment + P-Shuffle / Contrastive | ImageNet-1k | Pre-train | 1.0 |
| AugMix + P-Shuffle / L2 | ImageNet-1k | Pre-train | 1.0 |
| AugMix + P-Rotate / L2 | ImageNet-1k | Pre-train | 1.0 |
| AugMix + P-Infill / L2 | ImageNet-1k | Pre-train | 1.0 |
| AugMix + P-Shuffle / Contrastive | ImageNet-1k | Pre-train | 1.0 |
| Rand-Augment + P-Shuffle / Uniform | ImageNet-21k | Pre-train | 0.5 |
| Rand-Augment + P-Shuffle / L2 | ImageNet-21k | Pre-train | 0.5 |
| Rand-Augment + P-Shuffle / Contrastive | ImageNet-21k | Pre-train | 0.5 |
| Rand-Augment + P-Rotate / Uniform | ImageNet-1k | Fine-tune | 0.5 |
| Rand-Augment + P-Infill / Uniform | ImageNet-1k | Fine-tune | 1.0 |
| AugMix + P-Rotate / Uniform | ImageNet-1k | Fine-tune | 1.0 |
| Rand-Augment + P-Shuffle / Uniform | ImageNet-21k | Fine-tune | 0.5 |

## B  Hyper-parameters in Patch-based Negative Augmentation

For the temperature $\tau$ used in contrastive loss, we consistently observe that $\tau = 0.5$ works better in pre-training stage and $\tau = 0.1$ works better in fine-tuning stage. Therefore, we keep this setting for all the models in our paper.

Since we sweep the coefficient $\lambda$ in Eqn. 1 from the set $\{0.5, 1.0, 1.5\}$, we observe that for most of the cases, $\lambda = 1.5$ works the best. In total we have 48 models using loss regularization on negative views in Table 1, Table 2, Table 3 and Table 8. We use $\lambda = 1.5$ for all of them except those listed in Table 6, where either $\lambda = 0.5$ or $\lambda = 1.0$ works better. Actually, we find our proposed negative augmentation is relatively robust to $\lambda$. Therefore, we suggest using $\lambda = 1.5$ if readers do not want to sweep for the best value for this hyperparameter.

In Table. 7, we display the hyperparameters in each patch-based transformation that we use for the reported results in this work. Our algorithms are generally insensitive to these parameters, and we use the same hyperparameter for all the settings investigated in this work.

Table 7: Hyperparameters in patch-based transformations.

| Image Size | Stage | Transformation | Hyperparameter |
|---|---|---|---|
| $224 \times 224$ | Pre-train | P-Shuffle | patch size = 32 |
| $224 \times 224$ | Pre-train | P-Rotate | patch size = 16 |
| $224 \times 224$ | Pre-train | P-Infill | replace rate = 15/49 |
| $384 \times 384$ | Fine-tune | P-Shuffle | patch size = 64 |
| $384 \times 384$ | Fine-tune | P-Rotate | patch size = 32 |
| $384 \times 384$ | Fine-tune | P-Infill | replace rate = 3/8 |

Table 8: Patch-based negative augmentation is complementary to "positive" data augmentation. Top-1 accuracy on ImageNet-1k (IN), ImageNet-A (IN-A), ImageNet-C (IN-C) and ImageNet-R (IN-R) of ViT-B/16 pretrained and fine-tuned on ImageNet-1k. Our proposed patch-based negative augmentation are applied to either Rand-Augment (Cubuk et al., 2020) or AugMix (Hendrycks et al., 2020b). We display the accuracy of five different corruption severities on ImageNet-C (IN-C).

| Model | IN | IN-A | IN-C | | | | | IN-R |
|---|---|---|---|---|---|---|---|---|
| | | | 1 | 2 | 3 | 4 | 5 | |
| Rand-Augment (Cubuk et al., 2020) | 79.1 | 7.2 | 70.4 | 63.7 | 57.9 | 48.2 | 36.1 | 23.0 |
| + P-Shuffle / Uniform | 79.3 | 7.7 | 71.0 | 64.4 | 59.0 | 49.5 | 37.3 | 23.4 |
| + P-Rotate / Uniform | 79.3 | 8.1 | 71.1 | 64.6 | 59.0 | 50.0 | 37.6 | 23.8 |
| + P-Infill / Uniform | 79.2 | 7.8 | 71.1 | 64.6 | 59.1 | 49.5 | 37.3 | 24.0 |
| + P-Shuffle / L2 | 78.9 | 7.5 | 70.5 | 63.9 | 58.3 | 48.6 | 36.6 | 22.6 |
| + P-Rotate / L2 | 79.1 | 7.9 | 71.1 | 64.8 | 59.5 | 50.1 | 37.8 | 23.8 |
| + P-Infill / L2 | 78.8 | 7.4 | 70.5 | 63.8 | 58.2 | 48.4 | 36.0 | 23.2 |
| + P-Shuffle / Contrastive | 79.7 | 8.9 | 72.2 | 65.9 | 60.6 | 51.2 | 38.9 | 24.7 |
| + P-Rotate / Contrastive | 79.9 | 9.4 | 72.4 | 66.3 | 61.2 | 52.1 | 40.1 | 25.4 |
| + P-Infill / Contrastive | 79.9 | 9.3 | 72.3 | 66.1 | 61.0 | 51.8 | 39.5 | 25.0 |
| AugMix (Hendrycks et al., 2020b) | 78.8 | 7.7 | 71.4 | 65.2 | 60.5 | 51.9 | 40.2 | 24.9 |
| + P-Shuffle / Uniform | 79.2 | 8.0 | 71.6 | 65.7 | 61.2 | 52.8 | 41.4 | 25.7 |
| + P-Rotate / Uniform | 79.1 | 8.2 | 71.7 | 65.7 | 61.1 | 52.7 | 41.4 | 25.7 |
| + P-Infill / Uniform | 79.3 | 8.3 | 71.9 | 65.8 | 61.1 | 52.4 | 40.8 | 25.7 |
| + P-Shuffle / L2 | 78.8 | 7.9 | 71.8 | 65.8 | 61.0 | 52.4 | 40.7 | 25.7 |
| + P-Rotate / L2 | 79.0 | 8.3 | 71.9 | 66.0 | 61.5 | 52.9 | 41.6 | 26.0 |
| + P-Infill / L2 | 79.0 | 7.9 | 71.8 | 65.8 | 61.3 | 52.7 | 41.0 | 25.6 |
| + P-Shuffle / Contrastive | 79.6 | 9.0 | 72.9 | 67.2 | 62.8 | 54.6 | 43.2 | 27.3 |
| + P-Rotate / Contrastive | 79.6 | 9.8 | 72.6 | 66.9 | 62.6 | 54.5 | 43.5 | 27.5 |
| + P-Infill / Contrastive | 79.6 | 9.9 | 72.9 | 67.4 | 63.0 | 54.8 | 43.4 | 27.3 |

## C EXTRA RELATED WORK

Our work is also related to contrastive learning (Wu et al., 2018; Hjelm et al., 2019; Oord et al., 2018; He et al., 2020; Tian et al., 2020). The increasing number of negative pairs has shown to be important for representation learning in self-supervised contrastive learning (Chen et al., 2020a), where different images serve as negative examples for each other, and supervised contrastive learning (Khosla et al., 2020), where images with different classes are used as negative examples. Unlike the traditional setting of representation learning, our proposed contrastive loss serves as a regularization term with patch-based negative augmentations as extra negative data points.

Table 9: Effect of patch-based negative augmentation in pre-training and fine-tuning stages. Top-1 accuracies of ViT-B/16 pretrained and fine-tuned on ImageNet-1k. Under 'Stage' we denote which training stage patch-based negative augmentation is used. The best result under each setting is highlighted in **bold**.

| Pre-train on ImageNet-1k | | | | | |
|---|---|---|---|---|---|
| Model | Stage | ImageNet-1k | ImageNet-A | ImageNet-C | ImageNet-R |
| Rand-Augment (Cubuk et al., 2020) | - | 79.1 | 7.2 | 55.2 | 23.0 |
| + P-Shuffle / Uniform | Fine-tune | 79.1 | 7.1 | 55.3 | 23.0 |
| + P-Shuffle / Uniform | Pre-train | 79.3 | 7.6 | 56.2 | **23.5** |
| + P-Shuffle / Uniform | Both | **79.3** | **7.7** | **56.2** | 23.4 |
| + P-Shuffle / Contrastive | Fine-tune | 79.5 | 7.6 | 56.2 | 23.7 |
| + P-Shuffle / Contrastive | Pre-train | 79.4 | 8.5 | 56.8 | 24.0 |
| + P-Shuffle / Contrastive | Both | **79.7** | **8.9** | **57.8** | **24.7** |
| Pre-train on ImageNet-21k | | | | | |
| Model | Stage | ImageNet-1k | ImageNet-A | ImageNet-C | ImageNet-R |
| Rand-Augment (Cubuk et al., 2020) | - | 84.4 | 28.7 | 67.2 | **38.7** |
| + P-Shuffle / L2 | Fine-tune | 84.5 | 29.4 | 67.9 | 39.0 |
| + P-Shuffle / L2 | Pre-train | 84.4 | **29.9** | 67.5 | 38.8 |
| + P-Shuffle / L2 | Both | **84.5** | 29.7 | **68.0** | **39.6** |
| + P-Shuffle / Contrastive | Fine-tune | 84.4 | 29.2 | 67.5 | **38.7** |
| + P-Shuffle / Contrastive | Pre-train | **84.6** | 29.9 | 67.7 | 38.5 |
| + P-Shuffle / Contrastive | Both | 84.3 | **30.8** | **68.1** | 38.6 |

Table 10: Effect of positive augmentation in pre-training and fine-tuning stages. Top-1 accuracies of ViT-B/16 pretrained on ImageNet-21k and fine-tuned on ImageNet-1k. Under 'Stage' we denote which training stage Rand-Augment (Cubuk et al., 2020) is used.

| Model | Stage | ImageNet-1k | ImageNet-A | ImageNet-C | ImageNet-R |
|---|---|---|---|---|---|
| Rand-Augment | Pre-train | 84.4 | 28.7 | 67.2 | 38.7 |
| Rand-Augment | Both | 84.4 | 29.1 | 67.0 | 38.4 |

## D  WHEN TO USE POSITIVE DATA AUGMENTATION

As Steiner et al. (2021) observed that traditional (positive) augmentation can slightly hurt the accuracy of ViT if applied to fine-tuning stage, we compare the accuracy of a ViT-B/16 when positive augmentation (e.g., Rand-Augment (Cubuk et al., 2020)) is only applied to pre-training stage as well as both stages. As shown in Table 10, fine-tuning without Rand-Augment achieves slightly better performance. In addition, we also provide the results in Table 11 where we apply positive data augmentation in both stages, our proposed negative augmentation are still complementary to positive ones.

## E  EFFECT OF NEGATIVE AUGMENTATION IN CONTRASTIVE LOSS

Since we consistently observe that contrastive loss regularization works the best across all the settings that we have studied, we want to further investigate the effect of our proposed negative augmentation in contrastive loss. To this end, we design a stronger baseline by "only" excluding the patch-based negative augmentation in the negative set. Specifically, we replace $\mathbb{Q} \equiv \tilde{\mathbb{B}} \cup \mathbb{B}\setminus\{x_i\}$ in Eqn. 4 with $\mathbb{Q} \equiv \mathbb{B}\setminus\{x_i\}$. We denote this stronger baseline as "Contrastive*" and display the comparison in Table 12. We can see that even if we add the patch-based negative augmentation on top of this stronger contrastive baseline, we can still achieve extra improvement across robustness benchmarks. This further supports the effectiveness of our proposed patch-based negative augmentation in improving models' robustness.

Table 11: Top-1 accuracies for ViT-B/16 pre-trained and fine-tuned on ImageNet-1k using Rand-Augment (Cubuk et al., 2020) or AugMix (Hendrycks et al., 2020b) in both pre-training and fine-tuning. The proposed negative augmentation is added on top of either positive augmentation. Patch-based negative augmentation is complementary to "positive" data augmentation.

| Model | ImageNet-1k | ImageNet-A | ImageNet-C | ImageNet-R |
|---|---|---|---|---|
| Rand-Augment (Cubuk et al., 2020) | 79.2 | 7.9 | 55.1 | 23.2 |
| + P-Shuffle / Uniform | 79.4 (+0.2) | 8.6 (+0.7) | 56.0 (+0.9) | 23.3 (+0.1) |
| + P-Shuffle / L2 | 79.3 (+0.1) | 8.2 (+0.3) | 55.5 (+0.4) | 22.6 (-0.6) |
| + P-Shuffle / Contrastive | 79.4 (+0.1) | 9.4 (+1.5) | 56.9 (+1.8) | 24.9 (+1.7) |
| AugMix (Hendrycks et al., 2020b) | 78.7 | 8.8 | 57.9 | 24.7 |
| + P-Shuffle / Uniform | 79.4 (+0.7) | 9.0 (+0.2) | 59.0 (+1.1) | 25.3 (+0.6) |
| + P-Shuffle / L2 | 78.9 (+0.2) | 8.6 (-0.2) | 58.5 (+0.6) | 25.5 (+0.8) |
| + P-Shuffle / Contrastive | 79.2 (+0.5) | 10.2 (+1.4) | 59.4 (+1.5) | 26.6 (+1.9) |

Table 12: Effect of patch-based negative augmentation in contrastive loss regularization. Top-1 accuracies of ViT-B/16 trained with or without patch-based negative augmentation.

| Pre-train on ImageNet-1k | | | | |
|---|---|---|---|---|
| Model | ImageNet-1k | ImageNet-A | ImageNet-C | ImageNet-R |
| ViT-B/16 + Contrastive* | 78.7 | 8.1 | 53.5 | 22.8 |
| ViT-B/16 + Shuffle / Contrastive | 78.9 | 8.2 | 54.1 | 23.2 |
| ViT-B/16 + P-Rotate / Contrastive | 78.9 | 8.6 | 54.1 | 23.6 |
| Rand-Augment + Contrastive* | 79.7 | 8.9 | 57.6 | 24.7 |
| Rand-Augment + P-Rotate / Contrastive | 79.9 | 9.4 | 58.4 | 25.4 |
| Rand-Augment + P-Infill / Contrastive | 79.9 | 9.3 | 57.9 | 25.0 |
| AugMix + Contrastive* | 79.6 | 9.0 | 59.8 | 27.2 |
| AugMix + P-Rotate / Contrastive | 79.6 | 9.8 | 60.0 | 27.5 |
| AugMix + P-Infill / Contrastive | 79.6 | 9.9 | 60.3 | 27.3 |
| Pre-train on ImageNet-1k | | | | |
| Model | ImageNet-1k | ImageNet-A | ImageNet-C | ImageNet-R |
| Rand-Augment + Contrastive* | 84.1 | 29.7 | 67.6 | 39.2 |
| Rand-Augment + P-Shuffle / Contrastive | 84.3 | 30.8 | 68.1 | 38.6 |

## F   VISUALIZATION OF PATCH-BASED TRANSFORMATIONS

We display more examples with patch-based transformations without cherry-picking in Figure 5, Figure 6 and Figure 7.

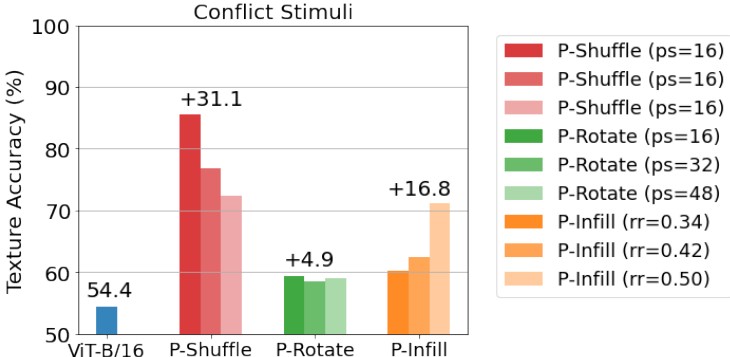

Figure 4: ViTs trained *only* on our patch-based transformations exhibit stronger texture bias. Each bar is the texture accuracy (%) on Conflict Stimuli (Geirhos et al., 2018), and a higher texture accuracy indicates the model has a higher bias towards texture. The "texture accuracy" is defined as the percentage of images that are classified as the "texture" label, provided the image is classified as either "texture" or "shape" label. The baseline model is ViT-B/16 in (Dosovitskiy et al., 2021) trained on original images. Other models are trained on patch-based transformed images, e.g., "P-Shuffle" stands for a ViT-B/16 model trained on patch-based shuffled images. Numbers above the bars are either accuracy (e.g., ViT-B/16) or the *max* accuracy difference between each model family and the baseline ViT-B/16. The patch size in P-Shuffle and P-Rotate and replacement ratio in P-Infill is denoted by "ps" and "rr" respectively.

Table 13: Patch-based negative augmentation effectively reduce models' texture bias on Conflict Stimuli (Geirhos et al., 2018). A higher texture accuracy indicates the model has a higher bias towards texture. The "texture accuracy" is defined as the percentage of images that are classified as the "texture" label, provided the image is classified as either "texture" or "shape" label.

| Pre-train on ImageNet-1k | | Pre-train on ImageNet-21k | |
|---|---|---|---|
| Model | Texture Accuracy | Model | Texture Accuracy |
| ViT-B/16 | 71.7 | Rand-Augment | 57.5 |
| + P-Rotate / Uniform | 66.5 | + P-Shuffle / Uniform | 56.4 |
| + P-Rotate / L2 | 67.2 | + P-Shuffle / L2 | 54.7 |
| + P-Rotate / Contrastive | 62.2 | + P-Shuffle / Contrastive | 56.4 |

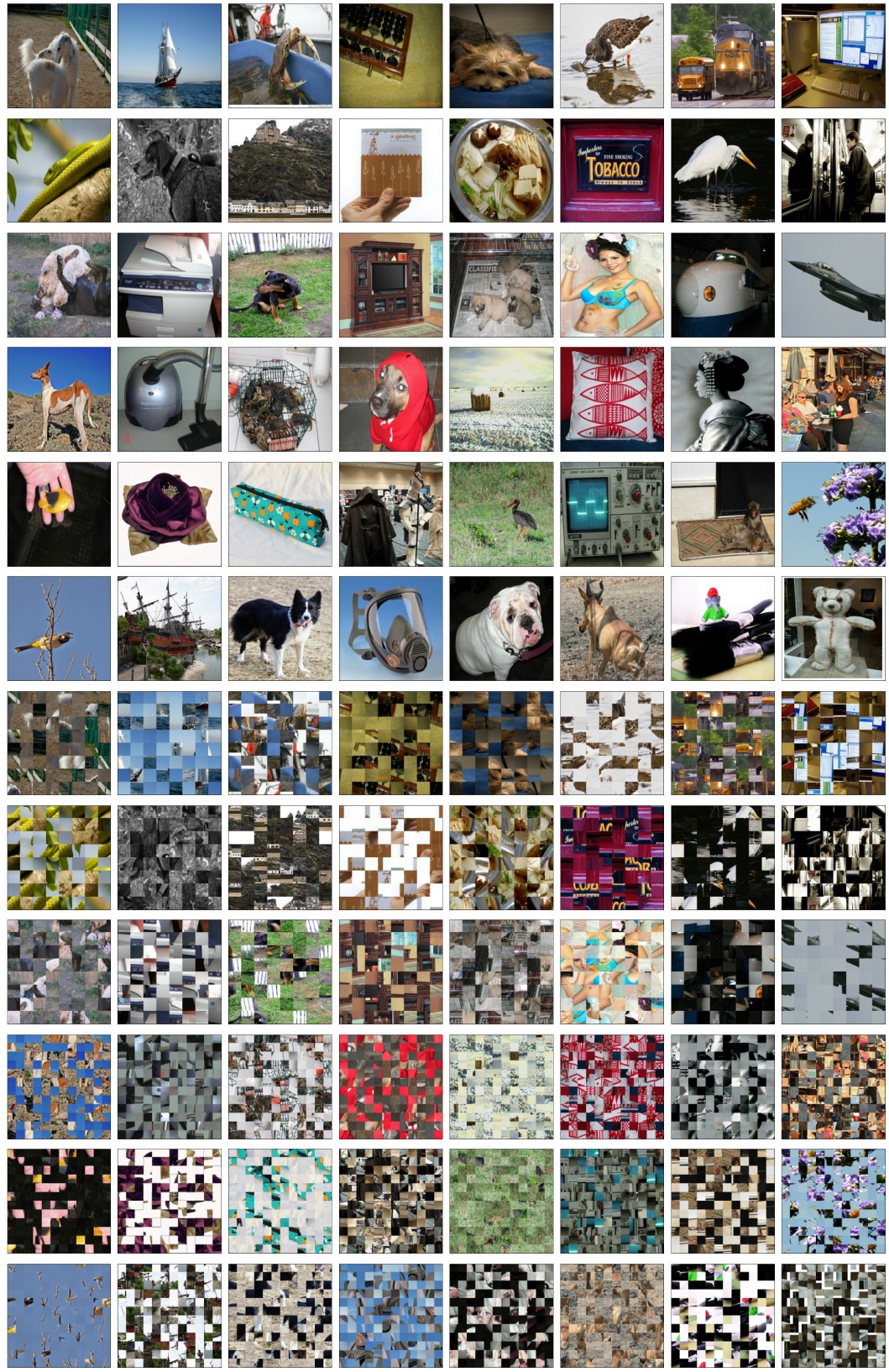

Figure 5: Examples of original images (on the top) and their corresponding patch-based shuffle (at the bottom) with either patch size 32 or 48 without cherry-picking.

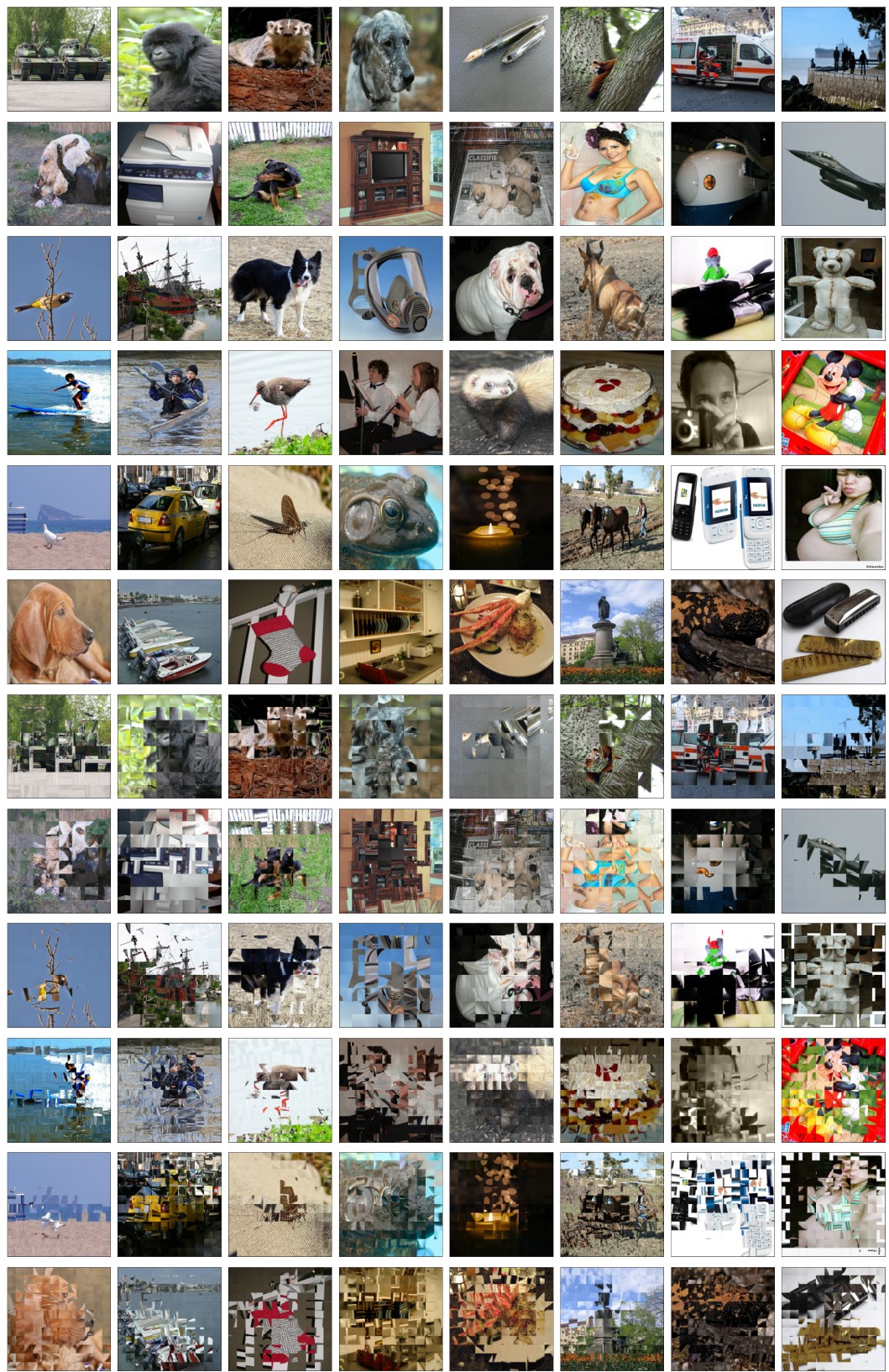

Figure 6: Examples of original images (on the top) and their corresponding patch-based rotation (at the bottom) with either patch size 32 or 48 without cherry-picking.

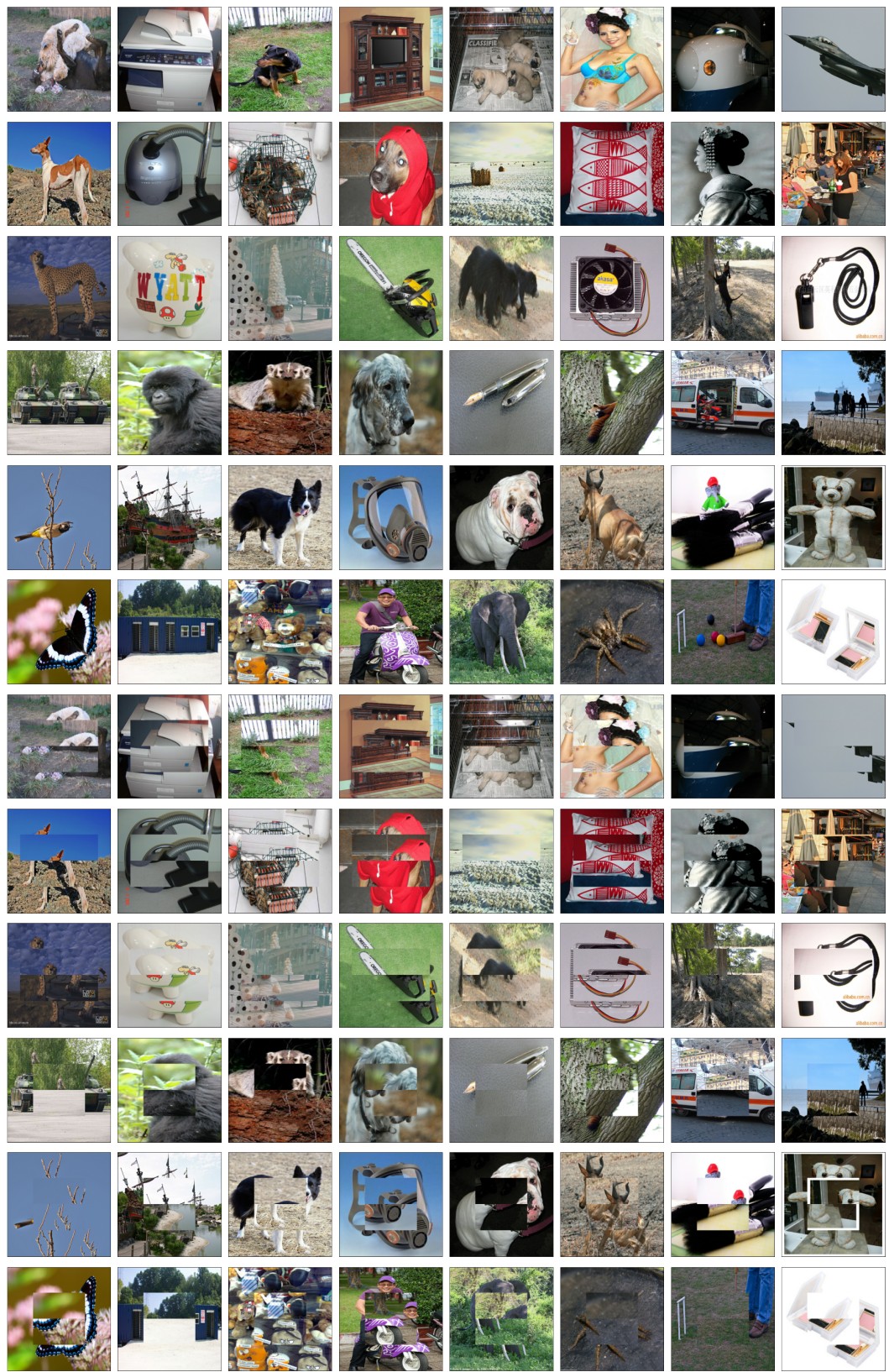

Figure 7: Examples of original images (on the top) and their corresponding patch-based infill (at the bottom) with either replace rate 0.25 or 0.375 without cherry-picking.

