# OpenReview forum: "Understanding and Improving Robustness of Vision Transformers through Patch-based Negative Augmentation"
_ICLR.cc/2022/Conference — ICLR 2022 Submitted_

### Official Review · Reviewer_j5h2 · 2021-11-01

**Correctness:** 4
**Technical Novelty And Significance:** 3
**Empirical Novelty And Significance:** 4
**Recommendation:** 5
**Confidence:** 4

**Main Review:**

**Strengths**

 - The stemming observations of this work is very interesting, where ViTs give stable predictions over those human-unrecognizable transformed samples. The author also provide extensive comparisons to support this findings.

 - While negative data augmentations have been explored in self-/unsupervised representation learning, they are rarely visited in vision transformers and from the robust perspective. Based on the paper motivation and experiment results, the proposed method looks promising.


**Weakness**

 - The incorporated loss function contains 3 individual components of a uniform loss, a L2 loss, as well as a contrastive loss. While comparisons in terms of accuracies are made, there exists no clear conclusion about the best choice or some detailed difference between them. In consequence, the message/contribution about the three losses is made uncertain. Further analysis or experiments that can support the statement in Sec 4 are expected.

 - Major experiments are conducted on the original ViT, or the ones with RandAug and AugMix. As has conveyed in the paper, the proposed negative augmentation is complementary to positive augmentations. Thus it's crucial to see the results where applying the proposed method to better models such as DeiT, since the training strategy adopted by DeiT is still ``positive''.

 - The transformer architecture used across most experiments is ViT-B/16, which uses a $16 \times 16$ token input. Will the transformer architectures that use a more fine-grained token representations, such as $8 \times 8$, produce dissimilar results? It's interesting to study the difference between them. Also, how do different scales of patch augmentation influence the results?

 - Recent progress in vision transformers cover modifications towards the tokenization process, where multi-scale or overlapping token embeddings are incorporated. Do you think the high confidence predictions towards unrecognizable images come from the non-overlapping token embedding in ViTs? If not, can such observations and augmentations apply to transformers architecture that uses overlapping token embeddings?

**Summary Of The Paper:**

This paper draws motivations from the observations that ViTs are insensitive towards patch-based transformations. Through detailed analysis by evaluating the models on such transformed inputs, the authors find a relatively stable accuracy, and term such features as **useful** but **non-robust** feature.  In order to push ViTs from these undesired features, patch-based negative augmentation and losses are proposed, which consistently improve the robustness of ViTs.

**Summary Of The Review:**

This paper provide some interesting findings on ViTs and propose patch-level negative augmentation. The authors also conduct experiments to validate the contribution of the augmentation and losses in details. However, the approach is only validated on the naive ViT-B/16 architecture and not convincingly enough. Moreover, additional analysis on losses and different architectures are expected.

---

> ### Author Response · Authors · 2021-11-23
> **Response to Reviewer j5h2**
>
> Thanks very much for your suggestions and we address your concerns below.
>
> **Q1: Which loss is the best?**
>
> A1: First, the main message that we want to emphasize in the paper is that we can use patch-based negative augmentation to capture non-robust features and then further use them as negative augmented views to improve the robustness of vision transformers. Therefore, we propose three different losses to show the effectiveness of patch-based negative augmentation as it works across all three losses. There is not a big difference between these three losses as the core ideas of them are the same: encouraging the difference between original images and their negative counterparts.
>
> Empirically we find that contrastive loss always works the best compared to uniform and L2 losses. We explain this as contrastive loss naturally incorporates the extra benefit of constraining the embeddings of positive pairs to be similar. We do have a more detailed discussion regarding the better performance of contrastive loss in **Appendix E**. Therefore, in practice, we would like to suggest readers to use contrastive loss due to its consistently better empirical performance. We will make it more clear in the final version.
>
> **Q2: Performance on DeiT.**
>
> A2: Since DeiT shares the **exact same architecture** as ViT and also relies on **Rand-Augment** for positive data augmentation, we believe our proposed patch-based negative augmentation should be able to generalize to DeiT as our proposed method has successfully improved the robustness of ViT with Rand-Augment. In addition, when ViT is pre-trained on ImageNet-21k, a much larger dataset, it already achieves state-of-the-performance and our method can still significantly improve its robustness further. Last, we also provide results below on ViT-B/32 as another architecture variant and our proposed method can also improve its robustness.
>
> **Q3. Change the patch size of the input image.**
>
> A3: The original ViT paper does not consider 8x8 image patches, which takes much longer training time. To address your concerns regarding the size of the input patch, we also perform experiments on ViT-B/32 which takes 32x32 input image patches. The experiments show that our method can still significantly improve the robustness.
>
> |Model           |                                               ImageNet-1k |  ImageNet-A | ImageNet-C | ImageNet-R |
> | :---          | :---       | :----   |          :--- |           :--- |
> |ViT-B/32     | 				72.5    |        3.7  	|  45.2    |         19.4	  |
> |+ P-shuffle / contrastive|		74.1 (+1.6)  | 3.9 (+0.2)|	  48.5 (+3.3) | 21.1(+1.7)   |
>
> **Q4: Change the patch size of patch-based transformation.**
>
> A4: We have already included an ablation study in **Section 5.5 in the paper** where we discussed the effect of different patch sizes of patch-based negative augmentation. We find that P-Shuffle and P-Rotate are insensitive to patch sizes from {16, 32, 48, 64, 96} for ViT-B/16, and P-Infill is robust to replace rates ranging from 1/3 to 1/2. The accuracy difference is smaller than 0.5% on ImageNet-1k as well as ImageNet-A and ImageNet-R.
>
> **Q5. Do you think the high confidence predictions towards unrecognizable images come from the non-overlapping token embedding in ViTs?**
>
> A5. As the first work to investigate the robustness of vision transformation to these patch-based transformations and found the features are useful but non-robust, we build our analysis on the original ViT architecture as it is a good starting point. Our intuition is that the high confidence prediction towards unrecognizable images is partially coming from the non-overlapping token embedding in ViT. In addition, we test ViTs’ accuracy on the patch-transformed images where the patch size of the patch-based transformations $ps_t$ is **not perfectly aligned** with the input patch size $ps_i$, e.g., $ps_t$ is not a multiple of  $ps_i$. This can approximate the effect of overlapped token embeddings.  Take P-Shuffle as an example (see results below), we find that when the patch size of the patch-based transformation ($ps_t$ = 24) and the input patch size ($ps_i $ = 16) are not perfectly aligned, the test accuracy of ViT on the unrecognizable images is lower compared to the case that the patch sizes are aligned. However, the test accuracy is still significantly higher than humans (a random guess is close to 0.1%) as well as comparable CNN-based network. Therefore, we conjecture that the high confidence is partially from the non-overlapping token embedding in ViTs.
>
> |Model |       patch size=24   |       patch size=32      |   ImageNet-1k clean   |
> | :---          | :---:        | :----:   |          :---: |
> |ViT-B/16       |  		32.9	|57.8   |     84.1|
> |BiT-ResNet101-3    |		9.2	| 24.6  |    84.0|

---

### Official Review · Reviewer_rziK · 2021-11-02

**Correctness:** 3
**Technical Novelty And Significance:** 4
**Empirical Novelty And Significance:** Not applicable
**Recommendation:** 6
**Confidence:** 5

**Details Of Ethics Concerns:**

None.

**Main Review:**

1. The exploration is novel: ViTs heavily use features that survived patch-based transformations but are generally not indicative of the semantic class to humans.
2. The experiments can support this paper’s claim and the designed patch-based negative augmentation is effective.


**Summary Of The Paper:**

This paper investigates the robustness of vision transformers. They first found that ViTs heavily use features that survived patch-based transformations but are generally not indicative of the semantic class to humans. However, these features are non-robust and the authors propose to use the images transformed with our patch-based operations as negatively augmented views and offer losses to regularize the training away from using non-robust features. Experiments on ImageNet show that patch-based negative augmentation consistently improves the robustness of ViTs.

**Summary Of The Review:**

My concerns are mainly focused on the experiments.

1. This paper only conducts experiments on the ViT while there are other SOTA vision transformer structures, e.g. swin-transformer [a]. The authors should conduct experiments with more transformer structures to demonstrate the generalization of the exploration and the proposed negative augmentation strategy.
[a] Swin transformer: Hierarchical vision transformer using shifted windows, ICCV2021

2. The authors only conduct experiments on one dataset, ImageNet. Although ImageNet is large enough, I wonder if the experiments’ results can be reproduced on the dataset of CIFAR-10-C and CIFAR-100-C which are utilized in AugMix.

3. The authors have tried with the transformation of patch-wise shuffle, rotation, and infill. How about other commonly utilized transformations in data augmentation? E.g., horizontal and vertical flip.

---

> ### Author Response · Authors · 2021-11-23
> **Response to Reviewer rziK**
>
> Thanks very much for your support of our work and we are happy to address your concerns below.
>
> **Q1: Other state-of-the-art vision transformers.**
>
> A1: We believe it is an interesting future direction to apply our proposed negative augmentation on other SOTA models. However, for our work, we believe it is a novel starting point with an inspiring observation that ViTs rely on useful but non-robust features preserved in small patches and we can improve models’ robustness through negative augmentation complementary to typical positive data augmentation. In addition, when we pre-train ViTs on ImageNet-21k with Rand-Augment, which is a very strong baseline model with state-of-the-art performance, we can still achieve robustness improvement.
>
> **Q2: Small-scale dataset.**
>
> A2: We implement our patch-based negative augmentation on CIFAR-100 and find that it can also significantly improve CIFAR-100-C datasets when we use patch-based shuffle on CIFAR-100 with uniform loss (only in the fine-tuning stage). The results are shown in Table 1.
>
> Table 1: P-Shuffle with uniform loss can improve robustness on CIFAR-100 and CIFAR-100-C when ViT-B/16 is fine-tuned on CIFAR-100.
>
> | Model          |   		CIFAR-100  |CIFAR100-C|
> | :---        |    :----:   |          :---: |
> |ViT-B/16 	|	    91.8  |           74.6|
> |+ P-Shuffle / Uniform  |    92.6 (+0.8)|	 77.0 (+2.4)|
>
> Note that if we apply our proposed negative data augmentation also in the pre-training stage, we can further improve the robustness, as discussed in Section 5.3 in the paper. Since ImageNet-1k is the large-scale dataset, which is the most challenging benchmark, the improvement of our method on smaller datasets such as CIFAR -100 is even **more significant**.
>
> **Q3: Patch-based horizontal and vertical flip.**
>
> A3: Thanks very much for your constructive suggestion and we test if patch-based horizontal and vertical flip transformations, another type of commonly utilized transformation, can also serve as negative augmented views. Experiments show that this new patch-based transformation can effectively improve robustness! Results are shown as below.
>
> |Model                  |                                        ImageNet-1k  | ImageNet-A | ImageNet-C | ImageNet-R|
> | :---        |    :----:   |          :---: | :---:       |   :----:   |
> |ViT + RandAug      | 				79.1      |       7.2  	|   55.2  	|23.8	|
> |+ P-Flip / contrastive	|		79.8 (+0.7) | 8.9 (+1.7)  | 57.7 (+2.5) 	|25.3 (+1.5)|
>
> This effectively supports our assumption that there is a big treasure of different types of negative augmentation, far beyond the three proposed patch-based transformations in the paper. Therefore, we believe our work, as the first to investigate this direction, can inspire more research to focus on designing and combing negative data augmentation to help improve robustness.

---

### Official Review · Reviewer_DY4w · 2021-11-02

**Correctness:** 4
**Technical Novelty And Significance:** 4
**Empirical Novelty And Significance:** 4
**Recommendation:** 8
**Confidence:** 4

**Main Review:**

Strength
------------
1. Thorough analysis of the hypothesis of patch-based robustness of ViTs and its detrimental effects on generalizability and robustness to image corruption (however, please see weaknesses). This provides further insight into properties of ViTs and complements existing work that investigates “positive” robustness of ViTs.
2. The proposed approaches on alleviating the detrimental effects are simple, intuitive and interesting.
3. Thorough investigation on approaches on how to alleviate this detrimental effect. These approaches are categorized as negative augmentations. The authors ablate the proposed method under a wide range of settings that provides insight and value to the research community.
4. Expanding on the previous point, the authors test negative augmentation in combination with positive augmentation and with larger datasets, both which may already solve the problem on their own but are shown to still improve in the presence of negative augmentation.
5. The paper is well-written and easy to read.

Weaknesses
-----------------
1. Experiments of Fig 2: This experiment is performed to demonstrate the robustness of ViTs towards patch-based transformations. This is done by running a trained ViT on ImageNet testing data on P-corrupted images and reporting the test accuracy. But for P-Shuffle and P-Infill, the accuracy of the ViTs clearly degrades (e.g ViT-B/32, P-Infill: ~80% -> 42%). Yet, these experiments of Fig 2 are used to empirically claim robustness. Whereas I tend to agree, I argue that a more non-subjective measure would be needed. I believe that repeating these experiments with a standard non-transformer CNN whose performance should degrade much faster would strengthen the argument being made for Fig 2.
2.  Experiments of Fig 3: This experiment is performed to emphasize features that are preserved by the patch-based transformations and to demonstrate that these aforementioned features are non-robust to out-of-distribution images. This is performed by training ViTs on P-corrupted images and evaluating them on non-corrupted OoD data. The resulting ViTs perform clearly worse and the authors use this as empirical evidence to support their hypothesis that the learned features are not generalizable. However, a degradation of performance is to be expected as the model has observed only P-corrupted images. Hence it is not clear whether the degradation comes from the non-robustness of the features or because of the shift in domain (from corrupted to non-corrupted). I believe that having an additional result, where the test data is corrupted in accordance with training data corruption would shed more light, as it would remove the parameter of domain shift (corrupted -> non-corrupted) from the equation. This would strengthen the author's argument.
3. All experiments seem to have been performed once and are being used to draw conclusions. As some improvements reside in the smaller ranges, I wonder how stable these results are when re-run. Hence, I would have liked to see some statistical significance testing, as the entire paper relies heavily on empirical results.
4. The authors propose three different corruption types and three different negative augmentation types. With all the tables and numbers, I’m missing a final conclusion on which combination is suggested by the authors to work the best and hence recommended.

Questions / Comments
--------------
1. For P-Rotate, is each patch rotated randomly by the same rotation degree or is a new rotation degree sampled for each?
2. Experiments of Fig 2: Are these re-trained by the authors or taken directly from Dosovitskiy et al? If the latter, it may be better to re-train them, as the experiments of Fig 3 use re-trained models to make sure minor differences in training code does not affect results
3. Do the experiments of Fig 2/3 use positive data augmentation and is the training set for both ImageNet (as opposed to ImageNet-1k)?
4. Out of curiosity, how would the authors extend the negative data augmentation to regression tasks?
5. Page 7: (...) and Steiner et al. (2021) have the similar observation -> have “made” similar observations
6. Have combinations of the P-corruptions been tried? E.g P-shuffle with P-rotate?


**Summary Of The Paper:**

This paper empirically investigates robustness properties of vision transformers (ViT) towards image augmentation strategies that destroy the semantic meaning of the image. The authors experimentally demonstrate that ViTs achieve high test set accuracies on ImageNet where the test data was transformed with patch-based transformations (I refer to these collectively as P-corrupted) which render the images unrecognizable to humans. They show that by emphasizing this property of ViT, it leads to detrimental effects to generalization to out-of-distribution (OoD) datasets.
As a remedy, the authors introduce several data augmentation techniques, termed negative augmentation, to alleviate ViTs tendencies to learn features that are robust to patch-based transformations
Extensive experimental results on ImageNet-(1k, A, C, R) ablate the proposed approach to extract the most effect hyperparameters and demonstrate the effectiveness of the proposed method

**Summary Of The Review:**

In summary, I enjoyed reading the paper. The authors perform thorough analysis of the hypothesis as well as their proposed approach. The weaknesses I mention are not minor and I would like to see them addressed. My biggest worry is the statistical analysis of the results, as  the entire paper relies heavily on experimentation.
Nevertheless, it did not dampen my positivity about the paper and therefore I feel comfortable recommending acceptance.

---

> ### Author Response · Authors · 2021-11-23
> **Response to Reviewer DY4w (Regarding Weakness)**
>
> Thanks very much for your support of our work and your suggestions are very constructive. We would like to address your concerns below and take your suggestions in our final version.
>
> **Q1: CNNs as a non-subjective measure in Figure 2.**
>
> A1: Thanks very much for your suggestion on using the performance of CNNs on patch-transformed images in Figure 2.  Taking P-Shuffle as an example, we report the performance of CNN based networks (ResNet101-3, ResNet50-1) which are also pre-trained on ImageNet-21k and then fine-tuned on ImageNet-1k for fair comparison. We can see that CNNs based networks degrade much faster, as shown in Table 1.
>
> Table 1: Test ViT and ResNet on patch-based shuffled images.
>
> |Model | patch size=32 |  patch size=48  |   patch size=64|ImageNet-1k clean  |
> | :--| :--:  | :--: | :---: | :---: |
> |ViT-B/16     |    57.8  | 66.9   |   71.5   |  84.1|
> |BiT-ResNet101-3  |   24.6   |  48.2  |     62.5   |  84.0|
>
> |Model |  patch size=32  |  patch size=64  |ImageNet-1k clean   |
> | :--    | :---:  |  :---: |  :---: |
> |ViT-B/32 |  51.0 |   66.5|  81.3|
> |BiT-ResNet50-1  | 13.5 | 55.3   |  79.5|
>
> **Q2: Adding one extra result which test the models on patch-transformed images without corruption in Figure 3?**
>
> A2: Thanks very much for your suggestion regarding Figure 3. If we understand correctly, you suggest that adding an extra result in Figure 3 with the test data transformed the same way as the training data. Take P-shuffle as an example, the model trained with P-shuffle have similar accuracy when tested on ImageNet-1k (79.1%) and P-shuffled ImageNet-1k (78.8%).
>
> To be more clear, we would like to illustrate more about Figure 3. All the comparisons are between a ViT model trained on clean images (blue bars in Figure 3) and ViT models trained on patch-transformed images (red, green orange bars in Figure 3). The difference between these two models are small on ImageNet-1k but are large on robustness benchmarks (ImageNet-A, ImageNet-C, ImageNet-R). Therefore, we conclude that features surviving small patches are useful features (small in-distributional accuracy difference between the two models) but are non-robust features (large robustness difference between the two models).
>
> **Q3: Statistical significance test.**
>
> A3: Since we run experiments on a wide range of different settings, e.g., 24 different models pre-trained on ImageNet-1k and 5 extra models pre-trained on ImageNet-21k, it is relatively challenging for us to report the standard deviation for each of them considering the large computing cost. Based on our experiments for sweeping the hyperparameters, the empirical results are stable (small standard deviation).
>
> To further address your concern, we choose one setting pre-trained on ImageNet-1k and ImageNet-21k respectively as an example and report the standard deviation as below. The reported number is computed over 4 independent runs. Mean$\pm$standard deviations are reported in Table 2.
>
> Table 2: Mean$\pm$standard deviations over 4 independent runs.
>
> | Model |  ImageNet-1k | ImageNet-A| ImageNet-C  |ImageNet-R  |
> | :- | :---: | :--:   | :--: | :--: |
> |**Pre-trained on ImageNet-1k**|
> |ViT-B/16  + RandAug|79.1$\pm$0.0|	7.3$\pm$0.1  |	55.3$\pm$0.1  |  22.9$\pm$0.2	|
> | + P-Rotate / contrastive|79.9$\pm$0.0	|9.2$\pm$0.1	|58.4$\pm$0.0  |  25.4$\pm$0.1	|
> |**Pre-trained on ImageNet-21k**|
> |ViT-B/16 + Rand-Augment | 84.4$\pm$0.0 |  28.7$\pm$ 0.2 | 67.2$\pm$0.0 | 38.7$\pm$0.1|
> |+P-Shuffle / L2  | 84.5$\pm$0.0 |   29.9$\pm$ 0.3 |   68.0$\pm$0.0 |    39.3$\pm$0.2|
>
> We can clearly see that patch-based negative augmentation consistently improves the robustness of models with statistical significance, complementary to typical data augmentation.
>
> **Q4: Final takeaway.**
>
> A4:  Our original intention was to highlight the importance and effectiveness of **patch-based negative augmentation** as it works across all three losses and three different patch-based transformations. To address your concern, we explain more here. In terms of three losses, contrastive loss always works the best compared to uniform and l2 losses as it naturally incorporates the extra benefit of encouraging the embeddings of positive pairs to be similar. In terms of three patch-based transformations, their performances are very similar to each other and it is hard to draw a conclusion which one is always the best. However, we believe there is a big treasure of different types of negative augmentation, far beyond these three transformations. For example, we take the suggestion from Reviewer rziK that using patch-based horizontal/vertical flip as another negative augmentation and experiments show that this patch-based transformation (P-Flip) can also improve robustness. Therefore, we believe our work, as the first to investigate this direction, can inspire more research to focus on designing and combing negative data augmentation to help improve robustness.
> We will include this analysis in the final version of our work.

---

> ### Author Response · Authors · 2021-11-23
> **Response to Reviewer DY4w (Regarding Questions / Comments)**
>
> We sincerely thank you for your support of our work and your constructive suggestions. Below we answer your questions.
>
> (1) For P-rotate, we randomly sample a new rotation degree sampled for each patch.
>
> (2) For ViT-B/16 used in both Figure 2 and Figure 3 and all the following experiments, we do use the retrained model and compare the result of ViT-B/16 achieved by us and the one reported in Dosovitskiy et al to make sure that they are the same. Actually, we use the exact same code base as Dosovitskiy et al. For other models in Figure 2, we used the models released by Dosovitskiy et al.
>
> (3) We do not use positive data augmentation in Figure 2 and 3 and they are both pre-trained on ImageNet-21k and then fine-tuned on ImageNet-1k. We do observe similar patterns if the model is pre-trained and fine-tuned on ImageNet-1k.
>
> (4) For regression, we believe we can still use L2 loss or contrastive loss for negative augmentation as they encourage the models to have different logits or representations for clean data and its negative counterpart. However, the uniform loss might not be easily generalized to regression models as it pre-defines the labels as uniform labels.
>
> (5) Thanks for the suggestion and we will change it to “have 'made' similar observations” in our final version.
>
> (6) We have not tried the combinations of different patch-based transformations and we believe it is an interesting future direction to study the combinations of different negative augmentations as there are many existing work proposed for effectively combining positive data augmentation to further improve models' performance.

---

### Official Review · Reviewer_FXi6 · 2021-11-04

**Correctness:** 1
**Technical Novelty And Significance:** 2
**Empirical Novelty And Significance:** 2
**Recommendation:** 3
**Confidence:** 4

**Main Review:**

Strengths:

+ The proposed idea is simple and easy to understand.

+ The illustrations and presentations are clear and easy to follow.

+ Most of the experiments are reasonable and the settings are clear.


Weaknesses:

- The paper concludes that the proposed method can significantly improve the out-of-distribution performance of ViTs. However, from the experimental results it is observed that the improvements led by the two losses, Uniform and L2, are limited. Only the contrastive loss leads to clear improvements to some extent, however,

- Contrastive loss is known to be very useful in improving the model training from the metric learning way, which helps to learn compact representations in class distributions in the learned representation feature space. Regarding this, the contribution of the contrastive loss is not clearly ablated in the proposed method. A fair baseline with contrastive loss should also be compared, that is, with only the clean images or positively augmented images for training, how about also appending the contrastive loss along with the classification loss, where in the contrastive loss all the anchors, positive examples, and negative examples are from the labeled clean images or positively augmented images?

- From Fig. 2 it is hard to conclude that ViTs are insensitive to patch transformations. The authors extends a lot on the range of the vertical axis to make the changes appeared to be narrow. However, from the figure it can still be observed that patch transformations significantly decrease the performance by up to more than 20%. This is hard to be understood as "insensitive".

- In Eq. (3) the loss is negative. Will it cause problems in optimization?

- Does the same finding also hold on CNNs?

**Summary Of The Paper:**

This paper presents that Visual Transformers (ViTs) are not sensitive to patch transformations (e.g. shift, rotation). It thus presents an idea that patch transformations can be used as negative augmentations to train the Visual Transformers (ViTs) to improve robustness similar to human perception. Three losses are thus applied to regularize the model training from being too confident to patch transformed images. A number of experiments on ImageNet-series datasets are presented to prove the finding of the authors.

**Summary Of The Review:**

I appreciate the authors' work from an interesting point of view. However, I think the statement of "ViTs are insensitive to patch transformations" does not hold, and the contribution of the contrastive loss is not clear. These two points make the current paper not reliable.

---

> ### Author Response · Authors · 2021-11-23
> **Response to Reviewer FXi6 (Questions 1-3)**
>
>
> **Q1: Limited improvement of uniform and L2 loss.**
>
> A1: In Table 1, we show the results of pre-training on ImageNet-21k with Rand-Augment, which is a strong baseline that already achieves state-of-the-art performance. When we use L2 loss for P-Shuffle, we can still consistently improve over all 3 robustness benchmarks. The result is an average over 4 independent runs (Mean $\pm$ standard deviation). Even on such a strong baseline, our proposed negative augmentation can still achieve 0.6 - 1.2% improvement on large-scale ImageNet datasets.
>
> Table 1: P-Shuffle with L2 loss can improve robustness when ViT-B/16 pre-trained on ImageNet-21k and then fine-tuned on ImageNet-1k.
>
>
> | Model |  ImageNet-1k    | ImageNet-A| ImageNet-C  |ImageNet-R  |
> | :---          | :---:        | :----:   |          :---: |           :---: |
> |ViT-B/16 + Rand-Augment | 84.4$\pm$0.0 |  28.7$\pm$ 0.2   |      67.2$\pm$0.0    |    38.7$\pm$0.1|
> |+P-Shuffle / L2      |              84.5$\pm$0.0       |   29.9$\pm$ 0.3     |   68.0$\pm$0.0   |    39.3$\pm$0.2|
>
> In addition, when we apply our proposed method on a small-scale dataset, e.g. CIFAR-100 and CIFAR-100-C, the improvement of robustness is **more significant**. Below we display the improvement of our negative augmentation using uniform loss on CIFAR-100-C, which includes 19 different types of corruptions with 5 different corruption levels. We can achieve 2.4% robustness improvement. Note that we applied patch-based transformation in the fine-tune stage only and it has already achieved a significant robustness improvement. If we apply our proposed negative data augmentation also in the pre-training stage, we can further improve the robustness, as discussed in Section 5.3 in the paper.
>
> Table 2: P-Shuffle with uniform loss can improve robustness on CIFAR-100 and CIFAR-100-C when ViT-B/16 is fine-tuned on CIFAR-100.
>
> |Model | CIFAR-100 | CIFAR-100-C|
> | :---          | :---       | :----   |
> |ViT-B/16 	|	    91.8      |              74.6|
> |+ P-Shuffle / Uniform   |92.6 (+0.8)       |  77.0 (+2.4)|
>
> **Q2: Discussion on contrastive loss.**
>
> A2: We do include a more detailed discussion regarding contrastive loss in Appendix E, where we discuss **the exact same** question as you mentioned. Compared to the stronger baseline which uses contrastive loss excluding our proposed negative augmentation (denoted as Contrastive*), we can still achieve improvement when we expand the negative pairs with our proposed negative augmentation. One example is shown in Table 3. We will move the discussion in **Appendix E** into the main paper and hope this addresses your concerns.
>
> Table 3:  Effect of patch-based negative augmentation in contrastive loss.
>
> | Model |  ImageNet-1k    | ImageNet-A| ImageNet-C  |ImageNet-R  |
> | :---          | :---:        | :----   |          :--- |           :--- |
> |ViT-B/16 + Contrastive*	|	78.7|	8.1		|53.5	|	22.8		|
> |ViT-B/16 + P-Rotate / Contrastive|	78.9 |	8.6(+0.5)	|54.1(+0.6)	|23.6(+0.8)    |
>
> **Q3: Insensitive to patch-based transformation**.
>
> A3: We agree that "insensitive" is a very subjective description and we use it as we compare the performance of vision transformers with humans who can hardly recognize the labels of these transformed images. Specifically, vision transformers exhibit a relatively less sensitive performance against those patch-based transformations compared to humans. We will be more cautious using the word ``insensitive’’ in the paper and make this more precise.
>
> In addition, compared to humans, we also test CNNs on the transformed images as a non-subject baseline. Taking P-shuffle as an example, we report the test accuracy using the original semantic classes as the ground-truth labels in Table 4. Both ViT and BiT are pre-trained on ImageNet-21k and fine-tuned on ImageNet-1k for fair comparison. We can see that there is a much more significant accuracy drop for convolutional based models such as BiT-ResNet101-3 and BiT-ResNet50-1. This further supports our claim that vision transformers are relatively less sensitive to these patch-based transformations.
>
> Table 4: Test ViT and ResNet on patch-based shuffled images.
>
> |Model |       patch size=32   |       patch size=48      |   patch size=64  |	ImageNet-1k clean   |
> | :---          | :---:        | :----:   |          :---: |           :---: |
> |ViT-B/16     |    57.8  |    66.9   |   71.5   |  84.1|
> |BiT-ResNet101-3  |   24.6   |  48.2  |     62.5   |  84.0|
>
> |Model |       patch size=32   |         patch size=64  |	ImageNet-1k clean   |
> | :---          | :---:        |          :---: |           :---: |
> |ViT-B/32     |    51.0     |   66.5	|  81.3|
> |BiT-ResNet50-1     | 13.5    |    55.3   |  79.5|

---

> ### Author Response · Authors · 2021-11-23
> **Response to Reviewer FXi6 (Questions 4-5)**
>
> **Q4: Any optimization problem of L2 loss?**
>
> A4: While we introduce L2 loss, we do mention in the paper that “Here the L2 distance is computed over the predicted probability rather than the logits $f(x; \theta)$ because empirically we observe that maximizing the difference of logits can cause numerical instability.” That is, we do observe an optimization problem if we use logits for L2 loss. However, we nicely address this problem by maximizing the difference of the predicted probabilities of clean examples and their negative counterparts because the predicted probabilities are constrained to be in [0, 1].
>
> **Q5: Does the same finding also hold on CNNs?**
>
> A5: As we illustrate in the response to Q3 above, when testing CNN based models on images transformed by patch-based shuffle, we do observe that there is a much more significant accuracy drop compared to vision transformers.

---

### Comment · Reviewer_DY4w · 2021-11-29
**Discussion on author feedback**

Dear AC, dear reviewers,

The authors have addressed my concerns in a satisfactory manner:

1. They have provided a study on standard CNNs greater vulnerability to patch-based image perturbation in comparison to ViTs. This makes the interpretation of ViTs being insensitive to patch-based losses less subjective and puts it better in perspective.
2. They convinced me of the sufficiency of the experiments of Fig 3. Although I would have still preferred the experiment I suggested, the explanation provided I deemed sufficient.
3. The standard-deviation on experiments was shown to be sufficiently low
4. A final-take away was provided. However, I would have preferred a more conclusive tone (e.g like the comment given to j5h2 to Q1).

I urge the authors to include these results into the next revision of the paper in order to improve it.

A point provided by other reviewers that concern me. Both reviewer rziK and j5h2 raise the valid point on how negative augmentation affect other transformer architectures (swin transformers, multi-scale, overlapping patches). If other SotA transformer architectures do not exhibit the same issue of invariance to patch-based image transformation, then perhaps it may be better to switch to those architectures instead of attempting to "fix" ViTs.

However, it is important to note that there is a limit to how much one can investigate in one paper. I argue that despite the prior shortcoming, there is sufficient enough new knowledge contained in this submission which is of value to the community. The authors achieve what they originally set out to do: Investigating and improving robustness of ViTs. Therefore I agree with the authors to leave further investigation on other architectures to future work.

In conclusion, I stand by my original rating of recommending acceptance but make a note that further investigation on other architectures is crucial to get a full picture on the improvements of negative augmentations.

---

### Decision · Program_Chairs · 2022-01-20

**Decision:**

Reject

**Comment:**

The reviews are split. The most significant concern seems to be the narrow focus of the paper: insensitivity of a very specific architecture, ViT to some patch-based transformations of the image. The paper aims to "understand and improve" the behavior of ViTs in this respect, but as the reviewers point out, the understanding (what exactly is the mechanism for this insensitivity) is lacking. Furthermore, there is a good reason to believe that other transformer architectures might not have a similar behavior. Ultimately both the lack of depth and the lack of breadth of the investigation suggest that the impact may be limited. I think this is not a good fit for ICLR.